# Molecular basis of interactions between CaMKII and α-actinin-2 that underlie dendritic spine enlargement

Ashton J Curtis[†], Jian Zhu[†], Christopher J Penny, Matthew G Gold*

Department of Neuroscience, Physiology and Pharmacology, University College London, London, United Kingdom

**Abstract** Ca²⁺/calmodulin-dependent protein kinase II (CaMKII) is essential for long-term potentiation (LTP) of excitatory synapses that is linked to learning and memory. In this study, we focused on understanding how interactions between CaMKIIα and the actin-crosslinking protein α-actinin-2 underlie long-lasting changes in dendritic spine architecture. We found that association of the two proteins was unexpectedly elevated within 2 minutes of NMDA receptor stimulation that triggers structural LTP in primary hippocampal neurons. Furthermore, disruption of interactions between the two proteins prevented the accumulation of enlarged mushroom-type dendritic spines following NMDA receptor activation. α-Actinin-2 binds to the regulatory segment of CaMKII. Calorimetry experiments, and a crystal structure of α-actinin-2 EF hands 3 and 4 in complex with the CaMKII regulatory segment, indicate that the regulatory segment of autoinhibited CaMKII is not fully accessible to α-actinin-2. Pull-down experiments show that occupation of the CaMKII substrate-binding groove by GluN2B markedly increases α-actinin-2 access to the CaMKII regulatory segment. Furthermore, in situ labelling experiments are consistent with the notion that recruitment of CaMKII to NMDA receptors contributes to elevated interactions between the kinase and α-actinin-2 during structural LTP. Overall, our study provides new mechanistic insight into the molecular basis of structural LTP and reveals an added layer of sophistication to the function of CaMKII.

*For correspondence:
m.gold@ucl.ac.uk

†These authors contributed equally to this work

Competing interest: The authors declare that no competing interests exist.

## Editor's evaluation

This fundamental study substantially advances our understanding of the synaptic targeting of a major postsynaptic protein kinase, CaMKII, that underlies the persistence of excitatory synaptic strength in synaptic plasticity. The evidence for the authors' claims is compelling, with cell biological, biochemical, as well as structural biological approaches. This work will be of broad interest to cell and computational biologists working on learning/memory.

## Introduction

Changes in synaptic connections between neurons are fundamental to learning and memory (*Citri and Malenka, 2008*). Calmodulin-dependent protein kinase II (CaMKII) plays a central role in long-term potentiation (LTP) of excitatory synapses following influxes of Ca²⁺ into postsynaptic spines (*Hell, 2014*). Activation of CaMKII by Ca²⁺/CaM leads to phosphorylation of postsynaptic proteins including AMPA-type glutamate receptors which has the overall effect of increasing postsynaptic responsiveness to glutamate release (*Anggono and Huganir, 2012*). CaMKII also serves a structural function in LTP (*Hojjati et al., 2007*) by nucleating networks of protein-protein interactions through its modular oligomeric structure (*Bhattacharyya et al., 2020*), consistent with its extremely high abundance in dendritic spines (*Erondu and Kennedy, 1985*). CaMKII holoenzymes form through oligomerisation of

the C-terminal hub domains of 12 subunits (*Chao et al., 2011*; *Myers et al., 2017*). The hub domains assemble into a two-tiered central ring from which the N-terminal kinase domains radiate (*Chao et al., 2011*; *Myers et al., 2017*). Binding of $Ca^{2+}$/CaM to a central regulatory segment releases the segment from the kinase domain enabling access to substrates and interaction partners (*Yasuda et al., 2022*). The CaMKII kinase domain has many documented substrates and binding partners (*Özden et al., 2020*). Formation of a highly stable complex between the CaMKII kinase domain and the C-terminal tail of NMDA receptor (NMDAR) GluN2B subunits is thought to be critically important for learning and memory (*Sanhueza et al., 2011*). Unlike the promiscuous kinase domain, the regulatory segment of CaMKIIα interacts only with α-actinins (*Dhavan et al., 2002*; *Walikonis et al., 2001*) besides CaM, while the sole binding partner of the hub domain is densin-180 (*Strack et al., 2000b*). Understanding how and when the regulatory and hub regions of CaMKII engage in these interactions is essential for a complete understanding of the structural role of CaMKII in LTP (*Hell, 2014*).

α-Actinins are structural proteins that form antiparallel rod-like dimers (*de Ribeiro et al., 2014*). In the heart, they are a key component of Z-discs where they crosslink actin filaments to titin (*Young et al., 1998*). In dendritic spines, α-actinins serve a more complex function with many additional binding partners including CaMKII, NMDARs (*Wyszynski et al., 1997*), PSD-95 (*Matt et al., 2018*), CaV1.2s (*Hall et al., 2013*), and densin-180 (*Walikonis et al., 2001*). Three of the four α-actinin isoforms have been detected in the postsynaptic density (*Walikonis et al., 2000*). The most abundant is α-actinin-2, which localises to dendritic spines of excitatory synapses (*Rao et al., 1998*; *Wyszynski et al., 1998*) where it interacts with the CaMKII regulatory segment independent of $Ca^{2+}$ through its third and fourth EF hands close to its C-terminus (gold, *Figure 1A*; *Jalan-Sakrikar et al., 2012*). Association with actinin only weakly stimulates the activity of CaMKII towards certain substrates (*Jalan-Sakrikar et al., 2012*; *Robison et al., 2005a*), therefore the interaction is thought to serve a structural role. Immunogold labelling shows that α-actinin-2 is present within the postsynaptic density (*Wyszynski et al., 1998*). Knockdown or overexpression of α-actinin-2 in cultured hippocampal neurons leads to defects in spine formation, with α-actinin-2 knockdown breaking the link between NMDAR activation and spine enlargement (*Hodges et al., 2014*) – a process known as structural LTP.

Given that α-actinin-2 is known to bind both CaMKII and actin filaments – and is required for spine enlargement – it would be logical if it bridged the two to support the structure of enlarged spine heads that accumulate following LTP. However, this model is at odds with the observation that $Ca^{2+}$/CaM (which activates CaMKII during LTP induction) competes with α-actinin-2 for binding to the CaMKII regulatory segment in vitro (*Jalan-Sakrikar et al., 2012*), undermining the notion that the two associate following CaMKII activation to bring about structural LTP. In this study, we have focused on reconciling these apparently contradictory observations. We employed in situ labelling in primary hippocampal neurons to reveal changes in association of CaMKIIα and α-actinin-2 following an NMDAR stimulation protocol that triggers structural LTP. We also examined the effect of disrupting the actinin-CaMKII interface on spine head enlargement triggered by NMDAR activation. We used isothermal titration calorimetry (ITC) and crystallography to determine whether the CaMKII kinase domain occludes access of α-actinin-2 to the regulatory segment in the inactive enzyme, and we have identified a potential mechanism for increasing access to the regulatory segment following induction of LTP. This combination of experiments has enabled us to put forward a more coherent molecular model of structural LTP.

## Results

### Association of CaMKII and α-actinin-2 is elevated following spine head enlargement

To investigate actinin association with CaMKII in neurons, we utilised proximity ligation assays (PLAs). We developed a set of vectors for mammalian expression of different fragments of α-actinin-2 (*Figure 1A*) in tandem with GFP. Each α-actinin-2 construct bears an N-terminal FLAG tag, so association with CaMKII can be monitored by PLA using paired anti-FLAG and anti-CaMKIIα antibodies. We transfected each construct in rat primary hippocampal neurons after 10 days in vitro (DIV10), and fixed the mature neurons on DIV14 for imaging anti-GFP immunofluorescence and α-actinin-2-CaMKIIα PLA puncta. For wild-type (WT) α-actinin-2, PLA puncta were visible in dendritic spines in unstimulated neurons (*Figure 1B*, arrow) at a density of 0.35±0.05 puncta per 10 µm dendrite (*Figure 1D*)

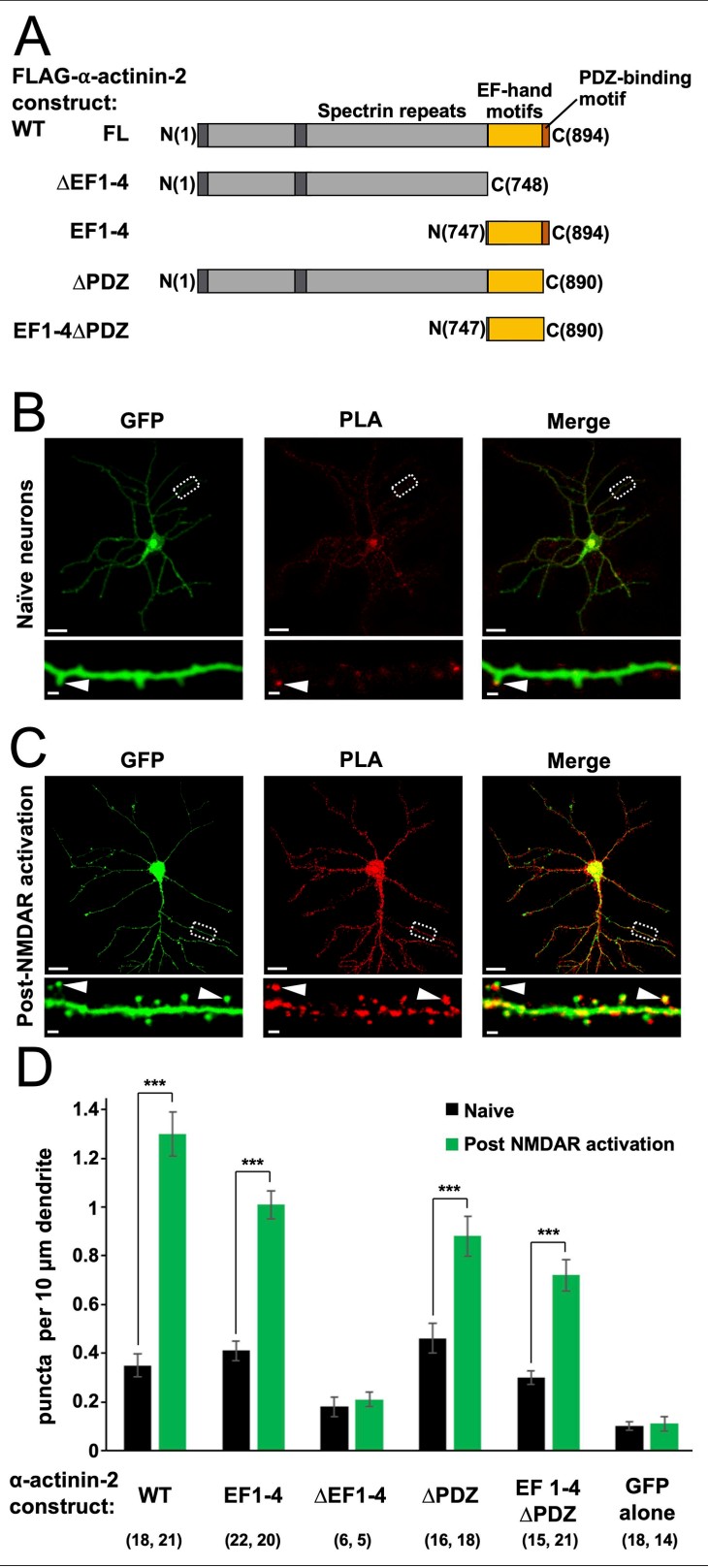

**Figure 1.** Changes in association of α-actinin-2 and calmodulin-dependent protein kinase II (CaMKII) following NMDA receptor (NMDAR) activation. (**A**) Topologies of α-actinin-2 constructs expressed in neurons. (**B**) and (**C**) show anti-GFP immunofluorescence (left column) and anti-CaMKIIα/anti-FLAG proximity ligation assay (PLA) puncta (middle column) in primary hippocampal neurons expressing FLAG-α-actinin-2 wild-type (WT) with GFP

*Figure 1 continued on next page*

*Figure 1 continued*

either before (**B**) or after (**C**) NMDAR activation. Scale bars are 20 µm (whole neuron images) and 1 µm (dendrite close-ups). (**D**) Quantitation of PLA puncta per 10 µm dendrite before (black) and after (green) NMDAR activation for the full range of α-actinin-2 constructs. Data are presented as the mean ± standard error (SE) and were collected from three independent cultures with the exception of ΔEF1–4 (one culture). The number of neurons analysed for each construct is shown in parentheses. Statistical comparisons were performed using unpaired two-tailed Student's t-tests (***p < 0.001).

The online version of this article includes the following source data and figure supplement(s) for figure 1:

**Source data 1.** Full proximity ligation assay (PLA) dataset.

**Figure supplement 1.** Proximity ligation assay (PLA) imaging of calmodulin-dependent protein kinase IIα (CaMKIIα) association with fragments of α-actinin-2.

**Figure supplement 2.** Frequency of FLAG-α-actinin-2/calmodulin-dependent protein kinase II (CaMKII) puncta by spine type.

**Figure supplement 2—source data 1.** Full dataset for anti-FLAG/anti-calmodulin-dependent protein kinase IIα (anti-CaMKIIα) proximity ligation assay (PLA) puncta frequency by spine type in neurons expressing FLAG-α-actinin-2.

– 3.6-fold higher (p=$2.2 \times 10^{-5}$) than in unstimulated neurons expressing GFP alone (***Figure 1—figure supplement 1A***). To investigate the notion that interactions between actinin and CaMKII stabilise the structure of potentiated spines, we employed a chemical model of LTP that uses glycine to activate NMDARs to trigger spine head enlargement (***Fortin et al., 2010***; ***Shahi and Baudry, 1993***). This is considered a relatively realistic chemical model for LTP (***Fortin et al., 2010***) that mimics other features including an increase in mini excitatory postsynaptic current amplitude (***Lu et al., 2001***) following stimulation. Anti-GFP immunofluorescence indicated that many mushroom-type spines had formed 4 hr after NMDAR activation, as expected (***Figure 1C***, left panel). This was associated with a 3.7-fold increase (p=$1.6 \times 10^{-10}$) in puncta in neurons expressing WT α-actinin-2 (***Figure 1C and D***). Bright puncta were localised to enlarged mushroom-type spines (***Figure 1C***, arrows): 44±7% of mushroom-type spines contained PLA puncta in naïve neurons, rising to 82±3% after NMDAR activation (***Figure 1—figure supplement 2***). In comparison, lower proportions of thin (19±5% pre-activation; 31±5% post-activation) and stubby (10±2% pre; 35±5% post) spines exhibited PLA puncta. In sum, this data is consistent with a role for α-actinin-2-CaMKIIα interactions in stabilising the enlarged structure of potentiated spines.

Next, we compared PLA puncta formation in neurons expressing different fragments of α-actinin-2 (***Figure 1A***). In line with evidence that α-actinin-2 EF hands mediate interactions with CaMKIIα (***Jalan-Sakrikar et al., 2012***; ***Robison et al., 2005b***), expression of a construct limited to the EF hand motifs ('EF1–4') yielded similar results to those obtained with full-length α-actinin-2 with a puncta density of 0.41±0.04 per 10 µm dendrite rising to 1.01±0.06 (p=$6.8 \times 10^{-11}$) following NMDAR activation (***Figure 1D***, ***Figure 1—figure supplement 1B***). Deletion of the EF hands ('ΔEF1–4') brought puncta density close to baseline levels irrespective of NMDAR activation with 0.18±0.04 and 0.21±0.03 puncta/10 µm dendrite before and after NDMAR activation, respectively (***Figure 1D***, ***Figure 1—figure supplement 1C***). Although PLA signal is a measure of protein proximity rather than direct protein-protein interaction, the low levels of puncta formation observed with the ΔEF1–4 construct indicate that puncta formation in this case is likely to correspond to direct interaction between CaMKII and α-actinin-2. The last four amino acids of α-actinin-2 (ESDL) are capable of binding to the PDZ domain of densin-180 (***Walikonis et al., 2001***) in vitro, and cooperative interactions between the three proteins in ternary complexes have been put forward as potentially important for supporting the structure of dendritic spines (***Robison et al., 2005b***; ***Walikonis et al., 2001***). Expression of FLAG-α-actinin-2 lacking the last four amino acids ('ΔPDZ') generated 0.46±0.06 PLA puncta per 10 µm in naïve synapses, rising to 0.88±0.08/10 µm (p=$2.9 \times 10^{-4}$) following NMDAR activation (***Figure 1D***, ***Figure 1—figure supplement 1D***). Similar responses were obtained with an EF hand construct lacking the PDZ motif ('EF1–4ΔPDZ') with 0.30±0.03 puncta per 10 µm dendrite rising to 0.72±0.07/10 µm (p=$1.1 \times 10^{-5}$) following NMDAR activation (***Figure 1D***, ***Figure 1—figure supplement 1E***). Overall, we saw a mild attenuation of puncta formation when the PDZ motif was absent, most apparent in the context of the full-length protein following NMDAR activation (1.48-fold reduction in puncta density, p=0.0017). Nevertheless, our data indicate that the actinin EF hands are sufficient for robust

α-actinin-2-CaMKII interactions, and this region underlies the marked elevation in interaction of the two proteins in enlarged spines.

## CaMKII-actinin interactions accumulate rapidly following NMDAR activation and are not affected by mutations that prevent regulatory phosphorylation

To understand how quickly CaMKII and α-actinin-2 associate following NMDAR activation, we fixed neurons expressing FLAG-α-actinin-2 at different times after the start of NMDAR activation then performed anti-FLAG/anti-CaMKIIα in situ PLA (*Figure 2—figure supplement 1*). PLA puncta accumulated rapidly reaching a plateau at 2 min following NMDAR activation (*Figure 2—figure supplement 1D*). Fitting the time series data to a Hill function reveals that CaMKII-α-actinin-2 interactions – according to in situ PLA – reach 50% maximal levels after 22±1 s (*Figure 2A*). This rate of accumulation is consistent with a recently proposed hypothetical time-scale of CaMKII signalling in dendritic spines following the induction of LTP (*Yasuda et al., 2022*).

CaMKII is subject to regulatory auto-phosphorylation at three sites (*Figure 2B*). Autophosphorylation at T286 supports autonomous low-level kinase activity (*Barcomb et al., 2014*). This form of CaMKII regulation is thought to be important in the initial induction but not maintenance of LTP (*Yasuda et al., 2022*). The CaMKII regulatory segment is also subject to negative feedback autophosphorylation at T305 and T306, with phosphorylation at either site preventing further activation by CaM (*Hanson and Schulman, 1992*; *Patton et al., 1990*). pT306 but not pT305 phosphorylation has been shown to reduce α-actinin-2 binding to CaMKII in vitro (*Jalan-Sakrikar et al., 2012*). We investigated the effect of preventing phosphorylation at the three sites on interactions between CaMKIIα and α-actinin-2 using PLA. We first expressed N-terminally V5-tagged WT CaMKIIα in tandem with FLAG-α-actinin-2 in primary hippocampal neurons. Anti-V5/Anti-FLAG PLA puncta were detected at the expected levels with 0.38±0.02 puncta per 10 μm dendrite in naïve neurons rising to 1.23±0.02 after NMDAR activation (*Figure 2C*, *Figure 2—figure supplement 2A*). PLA puncta remained at baseline levels irrespective of NMDAR activation if either construct was expressed in isolation (*Figure 2C*, *Figure 2—figure supplement 2E and F*). Puncta levels were little changed by alanine substitutions at either position 286 (0.40±0.01 puncta per 10 μm dendrite in naïve neurons rising to 1.17±0.01 after NMDAR activation, *Figure 2—figure supplement 2B*), position 305 (0.32±0.01 rising to 1.12±0.02, *Figure 2—figure supplement 2C*), or position 306 (0.44±0.01 rising to 1.24±0.02, *Figure 2—figure supplement 2D*). The absence of any marked effect when substituting T286A indicates that CaMKIIα auto-phosphorylation at T286 is not required for baseline or elevated association between CaMKIIα and α-actinin-2 in our system. It should be noted that T286 auto-phosphorylation is thought to be particularly important in the initial induction phase of LTP following summation of multiple $Ca^{2+}$ transients (*Yasuda et al., 2022*), which is not a function that would be required using our chemical LTP protocol. The absence of substantial changes compared to WT in either the T305A and T306A variants of CaMKIIα (*Figure 2C*) suggests that phosphorylation at either site does not play a significant role in regulating CaMKIIα-α-actinin-2 interactions. Overall, our data indicate that CaMKIIα-α-actinin-2 interactions increase within the first 2 min of LTP induction, and CaMKII phosphorylation is not essential for regulating interaction between the two proteins.

## Disruption of CaMKII-actinin interaction prevents spine enlargement following NMDAR activation

We noticed during PLA imaging that dendrites of neurons expressing α-actinin-2 EF1–4 seemed to exhibit fewer mushroom-type spines following NMDAR activation (*Figure 1—figure supplement 1B*) than those expressing either GFP alone (*Figure 1—figure supplement 1A*) or the full-length α-actinin-2 construct (*Figure 1C*). The α-actinin-2 EF1–4 construct (*Figure 1A*) lacks elements including the actin-binding domain and therefore can be expected to operate as a disruptor of actinin-mediated linking of CaMKII to the actin cytoskeleton. Furthermore, the only known interaction partner for the EF1–4 region of α-actinin-2 besides CaMKII is the muscle-specific protein titin (*Young et al., 1998*), so any effects of EF1–4 in neurons are likely to reflect destabilisation of native interactions between CaMKII and α-actinin-2. To investigate this disruptor effect further, we compared numbers of stubby (red), thin (amber), and mature mushroom-type (green) spines in neurons expressing either GFP alone (*Figure 3A*), GFP with α-actinin-2 EF1–4 WT (*Figure 3B*), or GFP with α-actinin-2 EF1–4 L854R (*Figure 3C*). The L854R mutation falls within

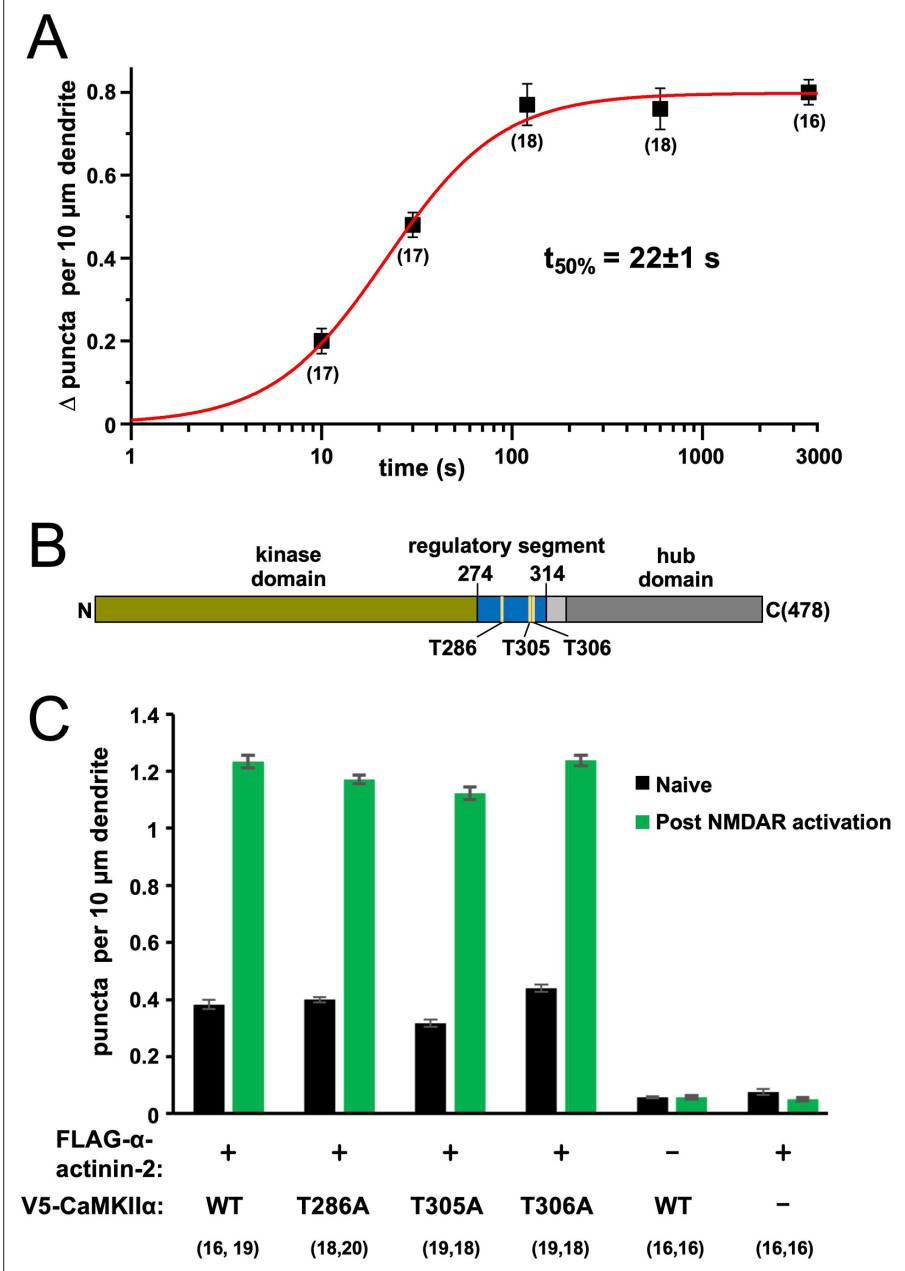

**Figure 2.** Time dependence and effect of phosphorylation site mutations on calmodulin-dependent protein kinase IIα (CaMKIIα)-α-actinin-2 association. (**A**) Plot showing change in anti-FLAG-α-actinin-2/CaMKIIα proximity ligation assay (PLA) puncta density in neurons fixed a different times following NMDA receptor (NMDAR) activation. The data were fit to a Hill function with max = 0.80±0.01, n=1.4±0.1, and $t_{50\%}$=22±1 s. (**B**) Topology of CaMKIIα showing positions of three regulatory threonines in the context of the kinase domain (green), regulatory segment (blue), linker (light grey), and hub domain (dark grey). (**C**) Quantitation of PLA puncta per 10 μm dendrite before (black) and after (green) NMDAR activation in neurons expressing different combinations of FLAG-α-actinin-2 and V5-CaMKIIα variants. For panels (A) and (C), data are presented as the mean ± SE, and the number of neurons analysed for each condition is shown in parentheses. Neurons were imaged deriving from three independent cultures for each condition.

The online version of this article includes the following source data and figure supplement(s) for figure 2:

**Source data 1.** Full proximity ligation assay (PLA) datasets for time course experiments, and experiments with calmodulin-dependent protein kinase IIα (CaMKIIα) phosphorylation site mutants.

*Figure 2 continued on next page*

*Figure 2 continued*

**Figure supplement 1.** Proximity ligation assay (PLA) imaging of calmodulin-dependent protein kinase IIα (CaMKIIα) association with α-actinin-2 PLA in neurons fixed at different time-points after NMDA receptor (NMDAR) activation.

**Figure supplement 2.** Proximity ligation assay (PLA) imaging of α-actinin-2 association with wild-type and phospho-null variants of calmodulin-dependent protein kinase IIα (CaMKIIα).

the fourth EF hand and disrupts α-actinin-2 binding to the CaMKII regulatory segment (*Jalan-Sakrikar et al., 2012*) thus serves as a negative control. PLA imaging confirmed that association with CaMKII was greatly reduced for the L854R variant (*Figure 3—figure supplement 1*). Prior to NMDAR activation, the total number of spines was approximately one-third lower in neurons expressing the WT EF1–4 construct than in the other two conditions (*Figure 3—figure supplement 2A*). For all three conditions, the distribution of spine types was initially similar (*Figure 3D*), with stubby spines predominating at ~1.5 stubby spines per 10 μm dendrite prior to NMDAR activation. However, whereas NMDAR activation triggered a transition from stubby to mushroom-type spines in neurons expressing GFP alone (*Figure 3A*) or GFP/EF1–4 L854R (*Figure 3C*), no such transformation occurred in neurons expressing the WT EF1–4 disruptor (*Figure 3B*).

Four hours after NMDAR activation (*Figure 3F*), stubby spines had decreased in abundance from 1.50±0.19 to 0.67±0.10/10 μm in GFP-only neurons (p=0.0013), and from 1.64±0.15 to 0.65±0.10/10 μm in neurons expressing L854R EF1–4 (p=7.2 × 10^{-6}). Furthermore, mushroom spines accounted for approximately half of all spines 4 hr after NMDAR activation (*Figure 3F*). In contrast, mushroom spines did not accumulate in neurons expressing WT EF1–4 disruptor, such that after 4 hr there were 7.2-fold fewer mushroom spines in these neurons than in GFP control neurons (p=3.43 × 10^{-11}) and 6.8-fold fewer than in L854R EF1–4 neurons (p=4.81 × 10^{-6}). The number of stubby spines also remained unchanged in WT EF1–4 neurons (1.24±0.16 before vs 1.22±0.18/10 μm 4 hr after NDMAR activation). The lack of plasticity in neurons expressing the WT EF1–4 disruptor was also reflected in analysis of changes in the average spine diameter upon NMDAR activation (*Figure 3—figure supplement 2B*). Mean spine diameter increased by 58.3% (p=7.1 × 10^{-8}) and 65.5% (p=1.3 × 10^{-4}) over 4 hr in GFP-only and L854R EF1–4 neurons, respectively, whereas expression of the WT EF1–4 disruptor limited the increase to only 11% (p=0.17). These results build on previous reports that siRNA-mediated knockdown of α-actinin-2 reduces mushroom-type spine formation following NMDAR activation (*Hodges et al., 2014*) by resolving a key role for the interface with CaMKII in this process. In sum, our PLA and spine imaging data indicate that α-actinin-2-CaMKII interactions accumulate upon induction of structural LTP via NMDAR activation, and that interaction between the two is necessary to support the formation of enlarged mushroom-type spines.

## The CaMKII kinase domain decreases affinity of α-actinin-2 for the regulatory segment

Previous studies have shown that EF hands 3 and 4 of α-actinin-2 (hereafter referred to as 'EF3–4') are sufficient for binding CaMKIIα via its regulatory segment (*Jalan-Sakrikar et al., 2012*; *Robison et al., 2005b*). Ca^{2+}/CaM binds to CaMKII by employing its four EF hands to wrap around the regulatory segment. Since the segment is partially buried against the kinase domain in the inactive kinase (*Chao et al., 2011*; *Rellos et al., 2010*), Ca^{2+}/CaM binds much more tightly to isolated regulatory segment than to constructs that include the kinase domain (*Forest et al., 2008*; *Rellos et al., 2010*). This is significant since factors that alter the accessibility of the regulatory segment, including T286 auto-phosphorylation (*Singla et al., 2001*) and association with NMDARs (*Bayer et al., 2001*), can trap CaM. For α-actinin-2, the assumption has been that EF3–4 is able to fully access the regulatory segment in inactive CaMKII since only two EF hands are involved in the interaction (*Jalan-Sakrikar et al., 2012*). However, our PLA data suggest that a mechanism exists for increasing CaMKIIα-actinin interactions following induction of LTP (*Figure 1D*). We therefore employed ITC using purified proteins (*Figure 4—figure supplement 1*) to determine whether the kinase domain impedes access of α-actinin-2 to the regulatory segment in inactive CaMKII. We compared binding to a peptide corresponding to the regulatory segment (positions 294–315), and to a longer construct (1–315) that includes the kinase domain. An N-terminal thioredoxin (Trx) tag was fused at the N-terminus of the longer construct – where it would not be expected to affect interactions with the regulatory segment

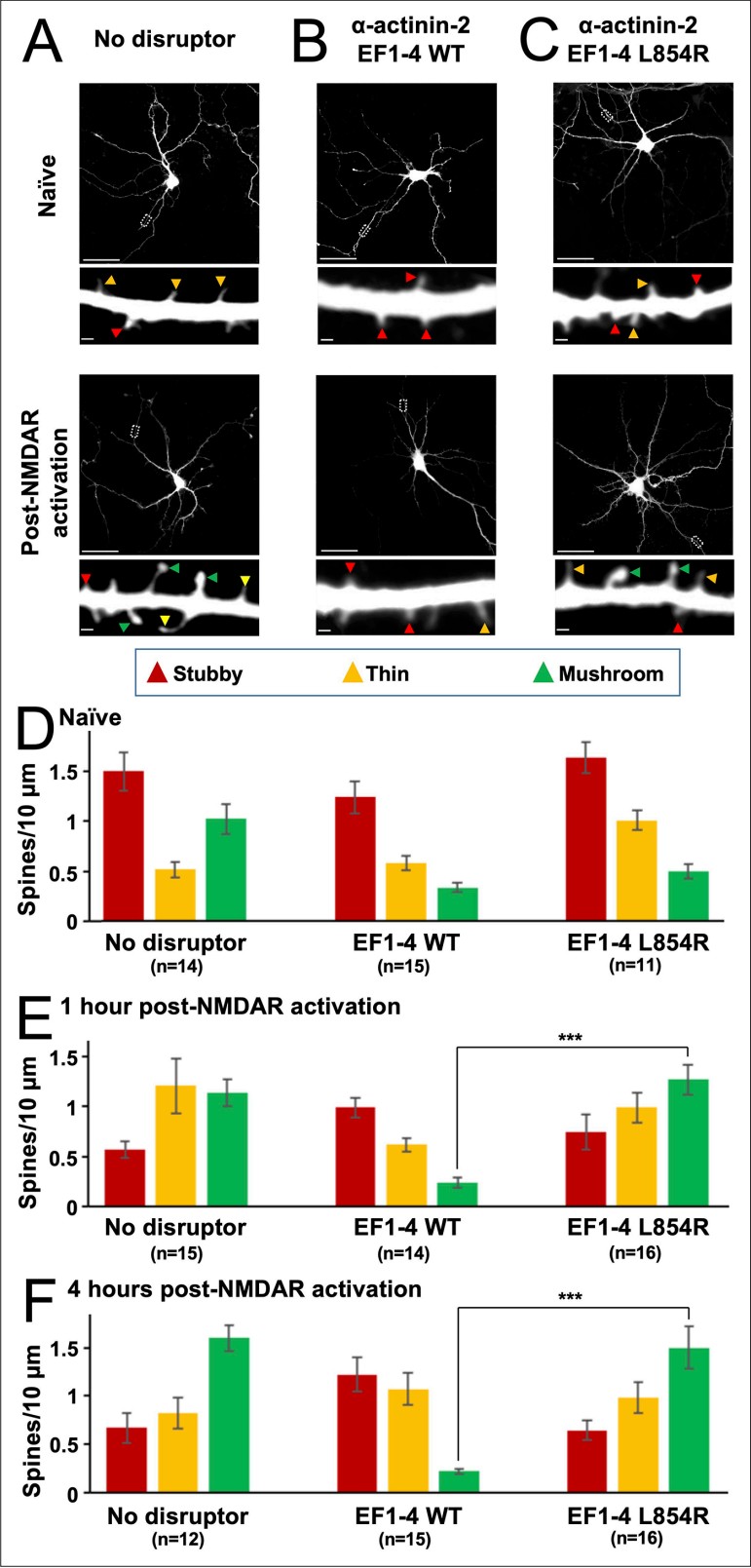

**Figure 3.** Effect of EF hand disruptors on changes in spine morphology following NMDA receptor (NMDAR) activation. Panels (**A–C**) show GFP imaging of primary hippocampal neurons transfected with either GFP alone (**A**), or GFP in combination with α-actinin-2 EF1–4 wild-type (WT) (**B**) or L854R (**C**). For (**A–C**), the upper rows show imaging of naïve synapses, and the lower rows show imaging 4 hr after NMDAR activation. Stubby (red),

*Figure 3 continued on next page*

*Figure 3 continued*

thin (orange), and mushroom (green) type synapses are highlighted with arrows. Scale bars correspond to 20 μm (whole neuron images) and 1 μm (dendrite close-ups). Panels (D–F) show quantification of spine types across the three conditions either before NMDAR activation (**D**), 1 hr after NMDAR activation (**E**), or 4 hr after activation (**F**). Data are presented as mean ± SE spines per 10 μm dendritic length. The number of neurons analysed for each condition is shown in parentheses. Neurons were imaged deriving from three independent cultures for each condition. Red, orange, and green bars indicate stubby, thin, and mushroom spine numbers, respectively. In panels E and F, statistical comparisons were performed using unpaired two-tailed Student's t-tests (***$p < 0.001$).

The online version of this article includes the following source data and figure supplement(s) for figure 3:

**Source data 1.** Full spine classification dataset.

**Figure supplement 1.** Proximity ligation assay (PLA) imaging of calmodulin-dependent protein kinase IIα (CaMKIIα) association with α-actinin-2 EF1–4 disruptors.

**Figure supplement 1—source data 1.** Full proximity ligation assay (PLA) dataset for comparison of wild-type (WT) and L854R disruptor variants.

**Figure supplement 2.** Effect of EF hand disruptors on total spine density and average spine diameter.

**Figure supplement 2—source data 1.** Full dataset for total spine number and changes in spine diameter for disruptor experiments.

(*Rellos et al., 2010*) – to ensure that the protein remained soluble at high concentrations necessary for ITC (*Costa et al., 2014*). EF3–4 bound to the regulatory segment peptide (294–315) with a dissociation constant ($K_d$)=32±1 μM (*Figure 4A*). In contrast, it was not possible to determine a $K_d$ for binding to the longer construct (1–315), with heat changes indistinguishable from background indicative of weaker binding (*Figure 4B*). Control experiments using CaM confirmed that, as expected, CaM binds more tightly to isolated regulatory segment ($K_d$ = 11±1 nM, *Figure 4C*) than to Trx-CaMKIIα 1–315 ($K_d$ = 2.8±0.2 μM, *Figure 4D*), consistent with previous studies (*Rellos et al., 2010*).

We next examined the notion that the first two EF hands of α-actinin-2 do not contribute to binding to the CaMKII regulatory segment (*Jalan-Sakrikar et al., 2012*; *Robison et al., 2005b*). Purified EF1–4 bound to the isolated regulatory segment (294–315) with $K_d$ = 11±0.2 μM (*Figure 4E*) – a threefold lower concentration than EF3–4 alone (*Figure 4A*). This is similar to $K_d$ = ~4 μM recorded for association of EF1–4 with peptide corresponding to titin Z repeat 7 (*Grison et al., 2017*). Like the shorter actinin construct (*Figure 4B*), no binding was detected between EF1–4 and Trx-CaMKIIα (1–315) (*Figure 4F*). Overall, our measurements show that – like CaM – α-actinin-2 is unable to fully access the CaMKII regulatory segment in the autoinhibited enzyme. Furthermore, our data reveal that when the regulatory segment is fully accessible, all four EF hands of α-actinin-2 are required for the highest affinity binding. Full thermodynamic parameters obtained for all ITC measurements are shown in *Table 1*.

## Structure of the core α-actinin-2-CaMKII interface

Previous structural models of α-actinin-2-CaMKII interaction have assumed that the third and fourth EF hands of α-actinin-2 bind to the CaMKII regulatory segment in a similar mode to the third and fourth EF hands of Ca$^{2+}$/calmodulin (*Jalan-Sakrikar et al., 2012*) in such a way that α-actinin-2 could fully access the regulatory segment in the inhibited kinase (*Jalan-Sakrikar et al., 2012*). However, our ITC measurements indicate that α-actinin-2 access to the regulatory segment is impeded in autoinhibited CaMKIIα (*Table 1*). Furthermore, PLA assays show increased association of the two proteins following chemical LTP protocols that activate CaMKII (*Figure 1*). To establish a structural basis for these findings, we crystallised α-actinin-2 EF3–4 in complex with a peptide corresponding to the CaMKIIα regulatory segment (positions 294–315). We solved the crystal structure at 1.28 Å resolution (*Figure 5—figure supplement 1A*) using X-ray diffraction with phasing through single-wavelength anomalous diffraction of native sulfur atoms (*Supplementary file 1*). The asymmetric unit contains two copies of the complex (*Figure 5—figure supplement 1B*) – the two copies are highly similar (RMSD 0.202 Å for all atoms, *Figure 5—figure supplement 1C*). Our analysis focuses on the first copy of the complex (chains A and B in PDB ID 6TS3) for which it was possible to position the full EF3–4 region (824–894) and positions 294–313 of the CaMKIIα regulatory segment in electron density.

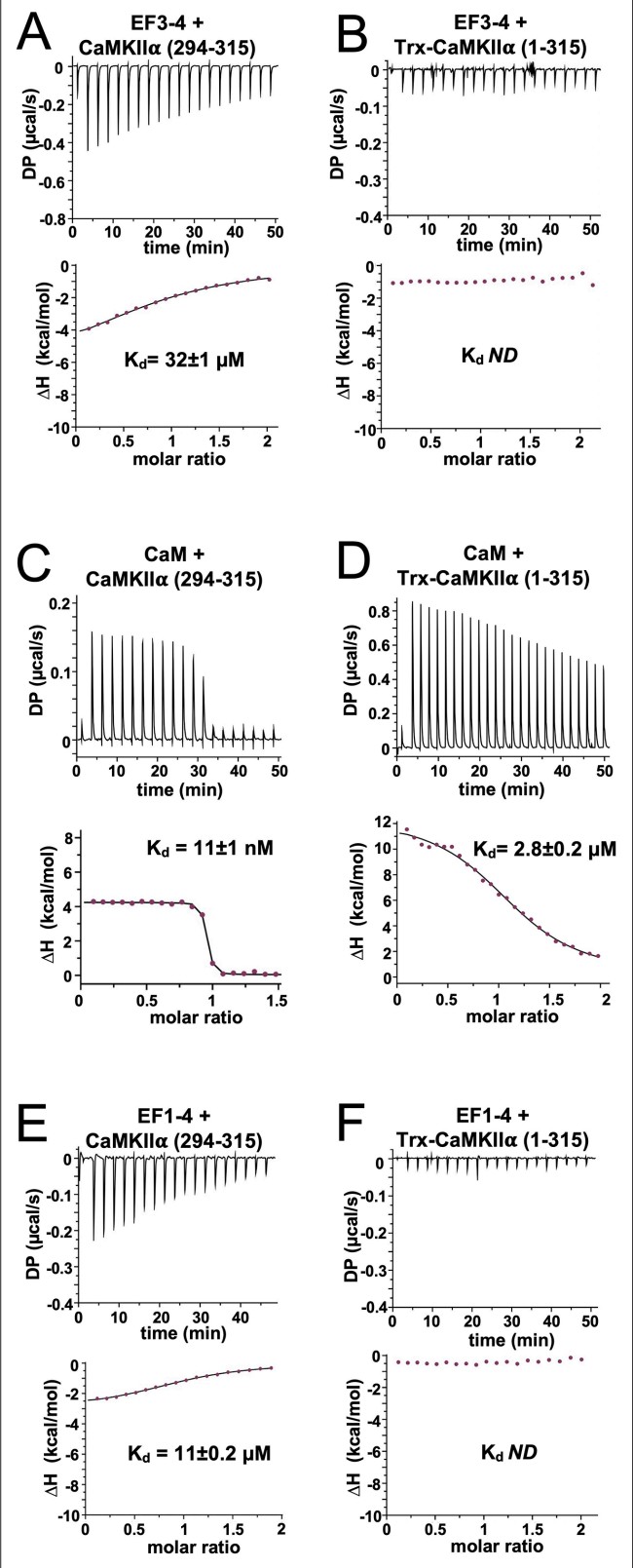

**Figure 4.** Isothermal titration calorimetry of interactions with the calmodulin-dependent protein kinase IIα (CaMKIIα) regulatory segment. Representative isotherms showing binding of α-actinin-2 EF3–4 to (**A**) peptide corresponding to CaMKIIα regulatory segment (294–315) and (**B**) a construct (1–315) corresponding to the kinase and regulatory segment regions of CaMKIIα. Binding of CaM and α-actinin-2 EF1–4 to the same two CaMKII

*Figure 4 continued on next page*

*Figure 4 continued*

regions are shown in panels (**C**) and (**D**) and (**E**) and (**F**), respectively. In all cases, the top sub-panels show the raw power output (µcal/s) per unit time; the bottom sub-panels show the integrated data including a line of best fit to a single site binding model. Stated $K_d$ values are averages from experimental replicates. ND = not determined.

The online version of this article includes the following source data and figure supplement(s) for figure 4:

**Source data 1.** Full isothermal titration calorimetry (ITC) dataset.

**Figure supplement 1.** Purified proteins.

**Figure supplement 1—source data 1.** Uncropped Coomassie-stained gels.

**Figure supplement 1—source data 2.** Raw image for panels (A–D).

**Figure supplement 1—source data 3.** Raw image for panel (E).

In the complex, the CaMKII regulatory segment (blue, *Figure 5A*) forms an amphipathic helix with four hydrophobic residues (L299, I303, M307, F313) aligned and engaged in van der Waals interactions with α-actinin-2 EF3–4 (orange, *Figure 5A*). M307$_{CaMKII}$ is buried closest to the centre of the EF3–4 domain, where it contacts the sidechains of EF3–4 residues F835 and L854. This binding mode is consistent with the reduction in PLA puncta that we observed with the EF3–4 mutation L854R (*Figure 3—figure supplement 1*) and with earlier pull-down experiments (*Jalan-Sakrikar et al., 2012*). α-Actinin-2 mutations S834R or Y861R have also been found to prevent association with CaMKII (*Jalan-Sakrikar et al., 2012*), and both of these residues also directly contact the regulatory segment in the crystal structure. The most N-terminal amino acid in the regulatory segment to engage directly with EF3–4 is L299$_{CaMKII}$, which interacts with Y861$_{EF3-4}$ (*Figure 5A*): unexpectedly, positions 294–298 are solvent exposed. At the other end of the regulatory segment, the benzene ring of F313$_{CaMKII}$ packs against the sidechain of I837$_{EF3-4}$ (*Figure 5B*). Deletion studies have found that the last nine amino acids (886–894) of α-actinin-2 are necessary for binding CaMKII (*Robison et al., 2005b*). This region includes Y889$_{EF3-4}$, which engages in van der Waals interactions with L304$_{CaMKII}$ through its benzene ring and H-bonds with the backbone oxygen of K300$_{CaMKII}$. The last four amino acids of α-actinin-2 ('ESDL') form a potential ligand for PDZ domains in proteins including densin-180 (*Robison et al., 2005b*; *Walikonis et al., 2001*). In the structure, G890$_{EF3-4}$ reorients the polypeptide chain away from the interface with CaMKII such that this PDZ ligand is accessible (*Figure 5B*). Mutational analyses have previously shown that phosphorylation at T306 but not T305 reduces binding to α-actinin-2 (*Jalan-Sakrikar et al., 2012*). This tallies with the crystal structure, in which T305 projects away from the interface with EF3–4 whereas the hydroxyl group of T306 engages in a H-bond network that includes Gln858$_{EF3-4}$, and its Cγ atom is packed against P885$_{EF3-4}$ (*Figure 5—figure supplement 1D*). In comparison, both T305 and T306 are buried in complex with CaM (*Figure 5—figure supplement 1E*). The EF3–4 region adopts a similar conformation in complex with the CaMKII regulatory segment as in the crystal structure of full-length apo α-actinin-2 (*de Ribeiro et al., 2014*) with 1.168 Å RMSD for all atoms. The CaMKIIα regulatory segment (blue, *Figure 5C*) occupies the same position as the neck region of α-actinin-2 (grey, *Figure 5C*). In neurons, association of the actin-binding domain

**Table 1.** Thermodynamic parameters for interactions between α-actinin-2 and calmodulin-dependent protein kinase IIα (CaMKIIα) constructs.

| Cell | Syringe | Reps | N | $K_d$ (µM) | ΔH (kcal/mol) | −TΔS (kcal/mol) |
|------|---------|------|---|-----------|---------------|------------------|
| EF3–4 | CaMKIIα 294–315 | 3 | 0.98±0.04 | 32±0.9 | −6.7±0.06 | 0.61±0.06 |
| CaMKIIα 1–315 | EF3–4 | 3 | ND | ND | ND | ND |
| EF1–4 | CaMKIIα 294–315 | 3 | 0.98±0.01 | 11±0.2 | −3.0±0.005 | −3.7±0.01 |
| CaMKIIα 1–315 | α-Actinin-2 EF1–4 | 3 | ND | ND | ND | ND |
| CaM | CaMKIIα 294–315 | 3 | 0.99±0.03 | 0.011±0.0007 | 4.3±0.2 | −14.6±0.2 |
| CaMKIIα 1–315 | CaM | 2 | 1.06±0.02 | 2.8±0.2 | 11.3±0.1 | −18.5±0.07 |
| EF3–4 | CaMKIIα 299–315 | 3 | 0.66±0.02 | 17.8±1.5 | −11±0.5 | 4.7±0.6 |
| CaM | CaMKIIα 299–315 | 3 | 1±0.01 | 0.047±0.001 | 4.51±0.04 | −14±0.05 |

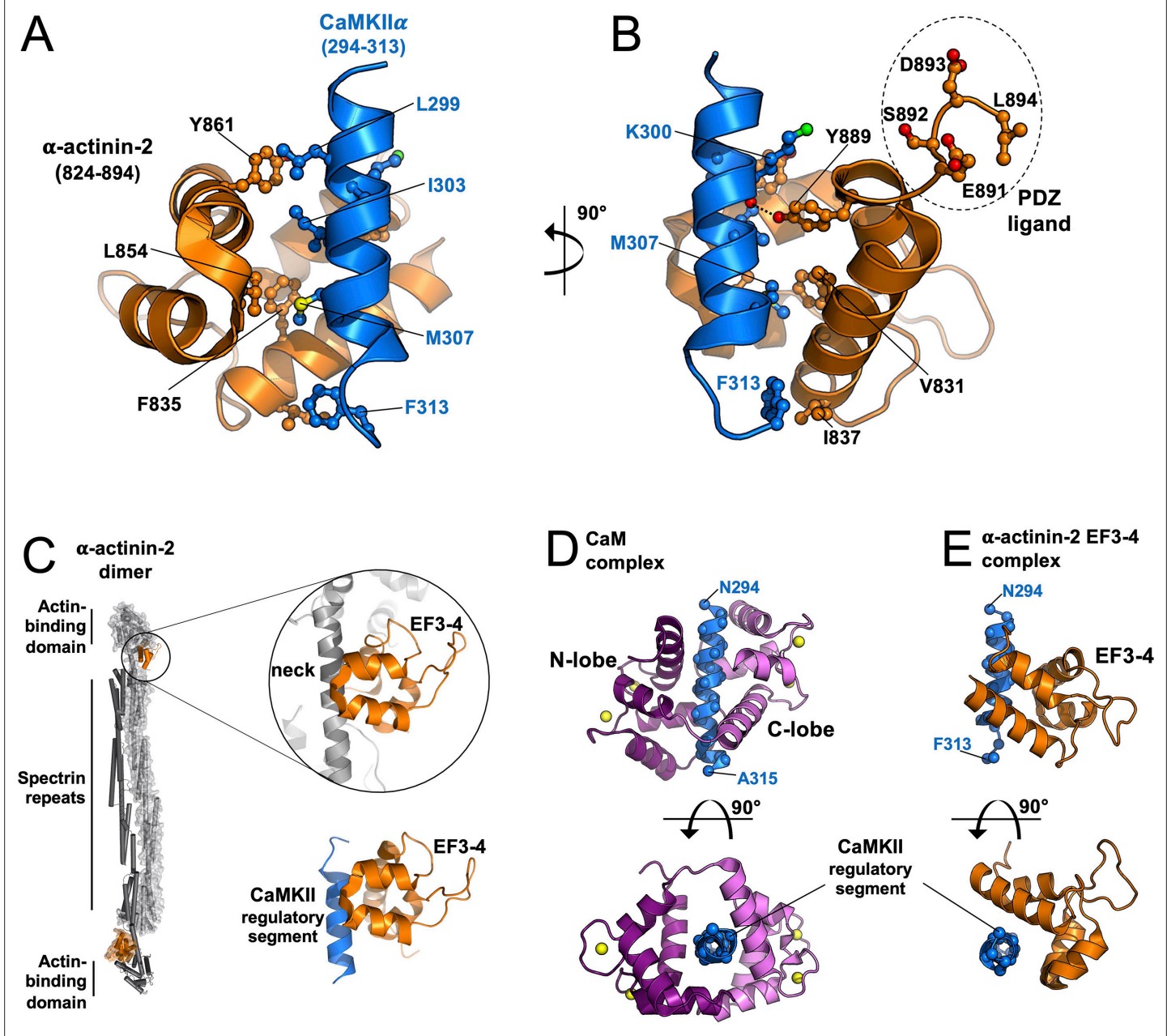

**Figure 5.** Structure of the core α-actinin-2-calmodulin-dependent protein kinase IIα (CaMKIIα) interface. Panels (**A**) and (**B**) show two views of the complex between α-actinin-2 EF3–4 (orange) and a peptide corresponding to the CaMKIIα regulatory segment (blue). The C-terminal tetrapeptide that is a ligand for PDZ domain-containing proteins including densin-180 is highlighted. (**C**) Comparison of α-actinin-2-EF3–4 domain association with the neck region (grey) in α-actinin-2 dimers (1H8B) and the CaMKII regulatory segment (6TS3). The structures were aligned through the EF3–4 domain. Panels (**D**) and (**E**) show two views comparing CaM (**D**) and α-actinin-2 EF3–4 (**E**) association with the CaMKIIα regulatory segment. The structures were aligned through the regulatory segment. For the CaM complex (2WEL), the N-lobe is coloured dark purple; the C-lobe is violet.

The online version of this article includes the following figure supplement(s) for figure 5:

**Figure supplement 1.** Additional features of the α-actinin-2 EF3–4-calmodulin-dependent protein kinase IIα (CaMKIIα) regulatory segment crystal structure.

**Figure supplement 2.** Calorimetry with calmodulin-dependent protein kinase IIα (CaMKII) 299–315 peptide.

of α-actinin-2 with actin filaments, and binding of PIP$_2$ phospholipids in the vicinity of the neck region, are thought to ensure that EF3–4 is available for interaction with CaMKII (*de Ribeiro et al., 2014*; *Hell, 2014*). The neck region residue Ile269 occupies the equivalent position to Met307$_{CaMKII}$ (*Figure 5A*), although the polypeptides run in opposite directions across the EF3–4 region.

Previous modelling of the CaMKII-actinin interaction has been built on the assumption that α-actinin-2 EF3–4 binds to CaMKII using the same binding mode as the CaM C-lobe (*Jalan-Sakrikar et al., 2012*). However, comparison to the crystal structure of CaMKIIδ (11–315) in complex with Ca$^{2+}$/CaM (*Rellos et al., 2010*) shows that the binding modes are distinct. Ca$^{2+}$/CaM fully encompasses the regulatory segment (whose sequence is identical between the α and δ isoforms) with the C-lobe (violet, *Figure 5D*) mediating most interactions to the hydrophobic face of the helix. The N-lobe (deep purple) is responsible for the bulk of H-bonding to the helical side that is solvent exposed in complex with α-actinin-2 EF3–4 (*Figure 5E*). The centre of mass of EF3–4 is rotated by ~50° relative to the CaM C-lobe when viewed along the central axis of the regulatory segment (lower panels, *Figure 5D and E*). In addition, the CaM C-lobe engages the regulatory segment approximately one helical turn further to its N-terminus than α-actinin-2 EF3–4, including interactions with positions A295 and K298. The conformation of the regulatory segment itself also differs between the two complexes. In complex with α-actinin-2 EF3–4, the helical structure breaks down at the C-terminal end to reorient F313 for interaction with I837$_{EF3-4}$ (light blue, *Figure 5—figure supplement 1F*). In complex with Ca$^{2+}$/CaM, alpha-helicity is maintained for the full length of the regulatory segment, which directs F313 in the opposite direction for interaction with the CaM N-lobe (dark blue, *Figure 5—figure supplement 1F*). Since CaMKIIα positions 294–298 are solvent exposed in the complex with α-actinin-2 EF3–4 but not CaM, we performed further ITC measurements with a truncated regulatory segment peptide (299–315) to corroborate the binding mode observed in the crystal structure. α-Actinin-2 EF3–4 bound CaMKIIα 299–315 peptide with K$_d$ = 17.8±1.5 µM (*Figure 5—figure supplement 2A*) – comparable to its affinity for CaMKIIα 294–315 (K$_d$ = 32±1). Consistent with the crystal structures, CaM associated with CaMKIIα (299–315) with K$_d$ = 47±1 nM (*Figure 5—figure supplement 2B*) – approximately five-fold higher than for the longer peptide (K$_d$ = 11 ± 1 nM). This also tallies with previous reports that positions 293–295 in the regulatory segment mediate interactions with CaM that markedly reduce the off-rate (*Putkey and Waxham, 1996*; *Waxham et al., 1998*). Overall, our structural data show how α-actinin-2 employs a unique binding mode to interact with the CaMKII regulatory segment.

## Occlusion of the regulatory segment to α-actinin-2 can be released by association with GluN2B

The reduced affinity of EF3–4/EF1–4 for the CaMKII regulatory segment in constructs that include the kinase domain (*Table 1*) suggests that the kinase domain impedes α-actinin-2 access to the regulatory segment. To understand the structural basis of this effect, we superimposed the EF3–4-regulatory segment complex structure on previously determined crystal structures of autoinhibited CaMKII. *Figure 6A and B* shows superimposition onto the structure of CaMKIIα (13–302) bound to the inhibitor indirubin (*Rellos et al., 2010*). In this structure L299 is the last visible residue in CaMKII (*Rellos et al., 2010*). The common residues 294–299 align closely with RMSD = 0.2 Å for backbone carbon and oxygen atoms (*Figure 6—figure supplement 1A*). The superimposition shows that – without reorientation or partial release of the regulatory segment from the kinase domain – both termini of the EF3–4 region will clash with the kinase domain (*Figure 6A and B*). Steric incompatibility of the kinase domain and EF3–4 region is most evident in the vicinity of the kinase domain αG helix (*Figure 6C*), which is located in the equivalent position to the last five amino acids of α-actinin-2. We also superimposed the EF3–4-regulatory segment complex onto the crystal structure of a chimeric form of full-length CaMKII, which incorporates WT sequence for CaMKIIα up to position T305 (*Chao et al., 2011*; *Figure 6—figure supplement 1B and C*). The two structures were aligned on the basis of the common positions 294–305. In this case, steric clashing was more pronounced since the regulatory segment kinks towards the kinase domain in this case (*Figure 6—figure supplement 1C*).

We found that α-actinin-2 association with CaMKII was markedly increased in dendritic spines 4 hr after chemical induction of LTP by NMDAR activation (*Figure 1*). Since our structural and calorimetry data show that the kinase domain occludes α-actinin-2 access to the regulatory segment in autoinhibited CaMKII, the mechanism underlying this increase in association is likely to involve regulatory segment release. Existing knowledge of CaMKII regulation during LTP suggests two possibilities:

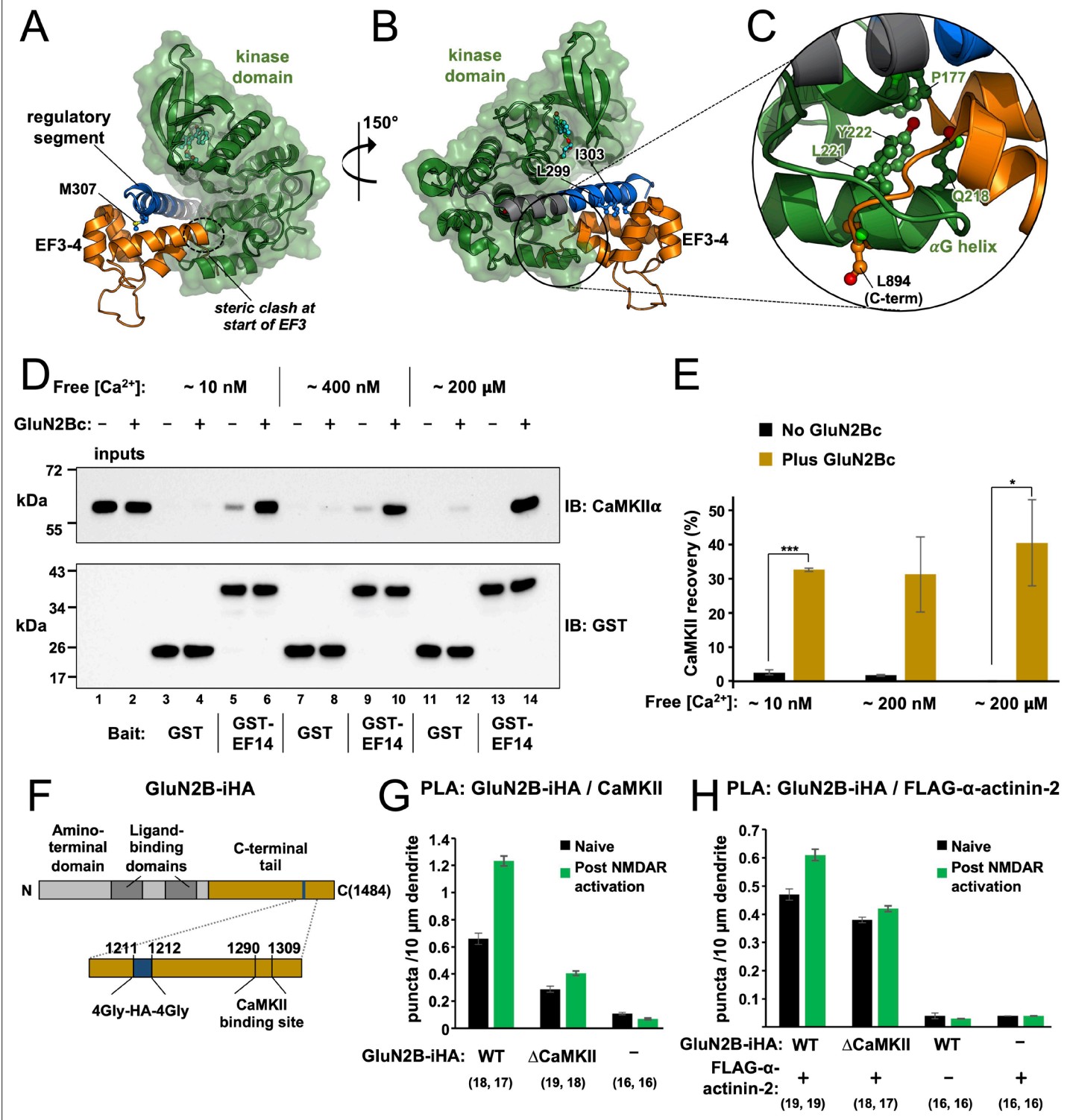

**Figure 6.** Effect of GluN2B on access to the regulatory segment. Panels (**A**) and (**B**) show two views of the superimposition of the α-actinin-2-regulatory segment complex (6TS3) over the structure of calmodulin-dependent protein kinase IIα (CaMKIIα) 1–299 (2VZ6). The structures were aligned using positions 294–299 of the regulatory segment. Regions of steric incompatibility between the kinase (green) and EF3–4 domain (orange) are highlighted. Panel (C) shows a close-up highlighting steric clashing in the vicinity of the CaMKIIαG helix. (**D**) Pull-down of purified CaMKIIα with magnetic beads charged with either GST or GST-EF1–4. CaMKII pull-down was compared ± GluN2Bc fragment, and at different final free $Ca^{2+}$ concentrations, as indicated. CaMKIIα and GST/GST-EF14 were detected by anti-CaMKIIα (upper) and anti-GST (lower) immunoblots (IBs). (**E**) Densitometry for pull-down experiments shown in the preceding panel showing CaMKII recovery at each free $Ca^{2+}$ concentration either with (gold) or without (black) GluN2Bc (n=3

*Figure 6 continued on next page*

*Figure 6 continued*

for all conditions). Statistical comparisons were performed using unpaired two-tailed Student's t-tests (*p < 0.05, ***p < 0.001). (**F**) Domain topology of GluN2B showing the location of the internal HA tag and CaMKII binding site within the C-terminal tail (gold). (**G**) Quantitation of anti-HA/anti-CaMKII proximity ligation assay (PLA) puncta per 10 µm dendrite before (black) and after (green) NMDA receptor (NMDAR) activation in neurons expressing GluN2B-iHA variants as indicated. (**H**) Quantitation of anti-HA/anti-FLAG PLA puncta per 10 µm dendrite before (black) and after (green) NMDAR activation in neurons expressing combinations of GluN2B-iHA variants and FLAG-α-actinin-2 as indicated. For panels (G) and (H), data are presented as the mean ± SE, and the number of neurons analysed for each condition is shown in parentheses. Neurons were imaged deriving from three independent cultures for each condition.

The online version of this article includes the following source data and figure supplement(s) for figure 6:

**Source data 1.** Densitometry breakdown for calmodulin-dependent protein kinase II (CaMKII) pull-down experiments and full proximity ligation assay (PLA) dataset for experiments investigating interactions with GluN2B.

**Source data 2.** Uncropped immunoblots.

**Source data 3.** Raw image for anti-calmodulin-dependent protein kinase II (anti-CaMKII) immunoblot.

**Source data 4.** Raw image for anti-GST immunoblot.

**Figure supplement 1.** Modelling relative orientations of kinase and EF3–4 domains relative to the regulatory segment.

**Figure supplement 2.** Proximity ligation assay (PLA) imaging of calmodulin-dependent protein kinase IIα (CaMKIIα) association with variants of GluN2B.

**Figure supplement 3.** Proximity ligation assay (PLA) imaging of α-actinin-2 association with GluN2B variants.

auto-phosphorylation at T286 and formation of NMDAR-CaMKII complexes (*Yasuda et al., 2022*). During the initial phase of LTP induction, CaMKII auto-phosphorylates at T286 to generate an autonomously active form with partial activity of ~20% (*Barcomb et al., 2014*). The pT286 modification is likely to partially disengage the regulatory segment from the kinase domain, however this modification does not endure for more than ~10 s following the induction of LTP (*Chang et al., 2017*; *Yasuda et al., 2022*). Furthermore, we show that CaMKIIα-α-actinin-2 interactions are not affected by the mutation T286A either before or after chemical LTP (*Figure 2C*). A more logical possibility to explain increased actinin-CaMKII association hours after LTP induction is the formation of NMDAR-CaMKII complexes that endure hours following CaMKII activation (*Bayer et al., 2001*; *Yasuda et al., 2022*). CaMKII binds tightly to a site centred on S1303 in the GluN2B C-terminal tail (*Omkumar et al., 1996*; *Strack et al., 2000a*), and recent crystallographic work shows how this sequence wraps around the kinase domain using a binding mode that necessitates full displacement of the regulatory segment from the kinase domain (*Özden et al., 2022*). To investigate the possibility that GluN2B supports CaMKIIα binding to α-actinin-2, we compared pull-down of full-length CaMKIIα with magnetic glutathione beads charged with either GST or GST-EF1–4 and determined the effect of including a purified fragment of the GluN2B tail spanning positions 1260–1492 (*Figure 4—figure supplement 1E*) – hereafter referred to as 'GluN2Bc'. In all cases, an initial pre-incubation step was included to enable CaMKII-GluN2B association (0.1 µM CaMKIIα, 3 µM CaM, 1.5 µM GluN2Bc incubated for 1 hr in buffer including 2 mM $CaCl_2$ and 0.5 mM ADP) prior to addition of EGTA and incubation with the charged magnetic beads. Reaction mixtures were supplemented with 2% BSA, and incubated with the magnetic beads for only 20 min to reduce basal pull-down in the absence of GluN2Bc. We also compared CaMKII pull-down with three final EGTA concentrations: 10, 2.5, or 1.8 mM corresponding to final approximate free $Ca^{2+}$ concentrations of 15 nM, 0.4 µM, and 200 µM (*Figure 6D*). CaMKII recovery without GluN2B was reduced from 2.5 ± 0.8% to 0 ± 0.1% (p=0.027, *Figure 6E*) in the absence of GluN2B moving from low to high final $Ca^{2+}$ (lanes 5 and 13, *Figure 6D*), consistent with previous reports that $Ca^{2+}$/CaM alone outcompetes α-actinin-2 for binding to CaMKII (*Robison et al., 2005a*). Some association without GluN2Bc at low $Ca^{2+}$ levels is consistent with our PLA imaging (*Figure 1D*) and with the original identification of the actinin-CaMKII interaction by the yeast two-hybrid method (; *Walikonis et al., 2001*) and it should be noted that this baseline interaction is more pronounced under less stringent binding conditions (*Jalan-Sakrikar et al., 2012*).

Addition of GluN2Bc led to striking increases in CaMKII recovery irrespective of EGTA concentration. At ~15 nM free $Ca^{2+}$, GluN2Bc increased recovery from 2.5±0.8% to 33 ± 0.5% (lanes 5 and 6, *Figure 6B*, p=5.0 × 10⁻⁶). At 0.4 µM $Ca^{2+}$, recovery increased from 1.7±0.3% to 31 ± 11% (lanes 9 and 10, p=0.055). Surprisingly, the effect was maintained at 200 µM free $Ca^{2+}$, with GluN2Bc increasing recovery from 0 ± 0.1% to 40 ± 13% (lanes 13 and 14, *Figure 6D*, p=0.032). The effects of GluN2Bc on CaMKII recovery are summarised in *Figure 6E*. Overall, these experiments indicate that association of

CaMKII with GluN2B subunits following LTP is a plausible mechanism to enable increased interaction between the kinase and α-actinin-2 in structural LTP. We next aimed to test this notion using in situ PLA measurements with GluN2B subunits. To this end, we modified a vector that expresses GFP-GluN2B by inserting an internal HA tag between positions G1211 and G1212 ('GluN2B-iHA', *Figure 6F*) We also generated a CaMKII binding-deficient variant of GluN2B-iHA ('ΔCaMKII') by introducing the double mutation L1298A/R1300Q (*Halt et al., 2012*). We first used PLA to measure association of GluN2B-iHA variants and endogenous CaMKII (*Figure 6—figure supplement 2*). Anti-HA/anti-CaMKIIα puncta were visible at 0.66±0.04 per 10 µm dendrite in neurons expressing WT GluN2B-iHA, rising to 1.23±0.04 (p=1 × 10$^{-11}$) following NMDAR activation whereas puncta were at baseline levels in the absence of GluN2B-iHA expression (*Figure 6G*). Puncta formation was suppressed in neurons expressing GluN2B-iHA ΔCaMKII with 0.29±0.02 puncta per 10 µm dendrite in naïve neurons rising to only 0.40±0.02 (p=3.7 × 10$^{-4}$) following NMDAR activation. Analysis of this data using two-way ANOVA supports an interaction between NMDAR activation and mutation of the CaMKII anchoring site (p=3.2 × 10$^{-10}$), consistent with the notion that CaMKII docks to this site during LTP.

Next, we measured anti-HA/anti-FLAG PLA puncta formation in neurons expressing GluN2B-iHA variants and FLAG-α-actinin-2 (*Figure 6—figure supplement 3*). Puncta formed at a frequency of 0.47±0.02 per 10 µm dendrite in naïve neurons expressing WT GluN2B-iHA and FLAG-α-actinin-2, rising 30 ± 6% to 0.61±0.02 following NMDAR activation (*Figure 6H*). Puncta formation was at baseline levels if either protein was expressed in isolation (*Figure 6H*). In neurons expressing GluN2B-iHA ΔCaMKII and FLAG-α-actinin-2, NMDAR activation only elevated puncta levels by 11 ± 5% from 0.38±0.01 to 0.42±0.01 per 10 µm dendrite (*Figure 6H*). Analysis of the data presented in *Figure 6H* by two-way ANOVA indicates an interaction between the presence of the CaMKII anchoring site and NMDAR activation (p=9 × 10$^{-3}$), which supports a mechanism in which CaMKII docking to GluN2B plays at least some role in supporting elevated CaMKII-α-actinin-2 association following structural LTP.

## Discussion

We propose a revised mechanism for spine enlargement in structural LTP (*Figure 7*) based on our findings. In naïve spines, the regulatory segment of CaMKII is sequestered by its kinase domain (*Figure 7A*) within inactive dodecamers, ensuring only limited association with α-actinin-2 in the ground state (*Figure 1D*). Upon induction of LTP by Ca$^{2+}$ influx through NMDARs, activated Ca$^{2+}$/CaM binds to the regulatory segment of CaMKII which exposes the substrate-binding groove of the kinase domain (*Figure 7B*). The activated kinase phosphorylates key substrates including AMPA receptors and forms a highly stable complex with NMDARs by interaction with a substrate-binding site within the C-tail of GluN2B subunits. The duration of CaMKII activation is prolonged beyond the duration of Ca$^{2+}$ elevation by auto-phosphorylation at T286, although pT286 autonomy is reversed by phosphatases including protein phosphatase 1 within ~10 s (*Yasuda et al., 2022*). In complex with NMDARs, the substrate-binding groove of the CaMKII kinase domain is occupied (*Özden et al., 2022*), which leaves the regulatory segment accessible to interact with α-actinin-2 (*Figure 7C*) enabling new interactions between the two to be established within the first 2 min of LTP induction. In this way, α-actinin-2 is enriched in dendritic spine heads where it supports spine head enlargement through its ability to crosslink actin filaments via its N-terminal actin-binding domain (*Figure 7C*). This updated mechanism fits with an influential 'synaptic tagging' theory (*Sanhueza and Lisman, 2013*), whereby long-lasting CaMKII-NMDAR complexes formed during LTP induction serve as primers for more global structural changes in dendritic spines that occur after the initial Ca$^{2+}$ signal has subsided.

The simplified model presented in *Figure 7* is broadly consistent with many additional protein-protein interactions involving CaMKII and α-actinins that are known to contribute to the positioning of the two proteins in dendritic spines. Densin-180 includes a C-terminal PDZ domain that binds to the PDZ ligand at the C-terminus of α-actinin-2 (*Walikonis et al., 2001*), and a motif ~100–150 amino acids from its C-terminus that associates with the CaMKII hub domain (*Strack et al., 2000b*). Earlier studies have shown that CaMKII can form tripartite complexes with densin-180, α-actinin-2, and GluN2B (*Robison et al., 2005b*). In the crystal structure (*Figure 5B*), the PDZ ligand at the C-terminus of EF3–4 is available for interaction which suggests that densin-180 could form a tripartite complex with CaMKII and α-actinin including a single CaMKII protomer. α-Actinin-2 is also known to interact with other NMDAR subunits and PSD-95. α-Actinin-2 binds to the C-tails of GluN1 subunits (*Wyszynski et al., 1997*) through its central spectrin repeats (*Figure 1A*), which could be compatible with binding to

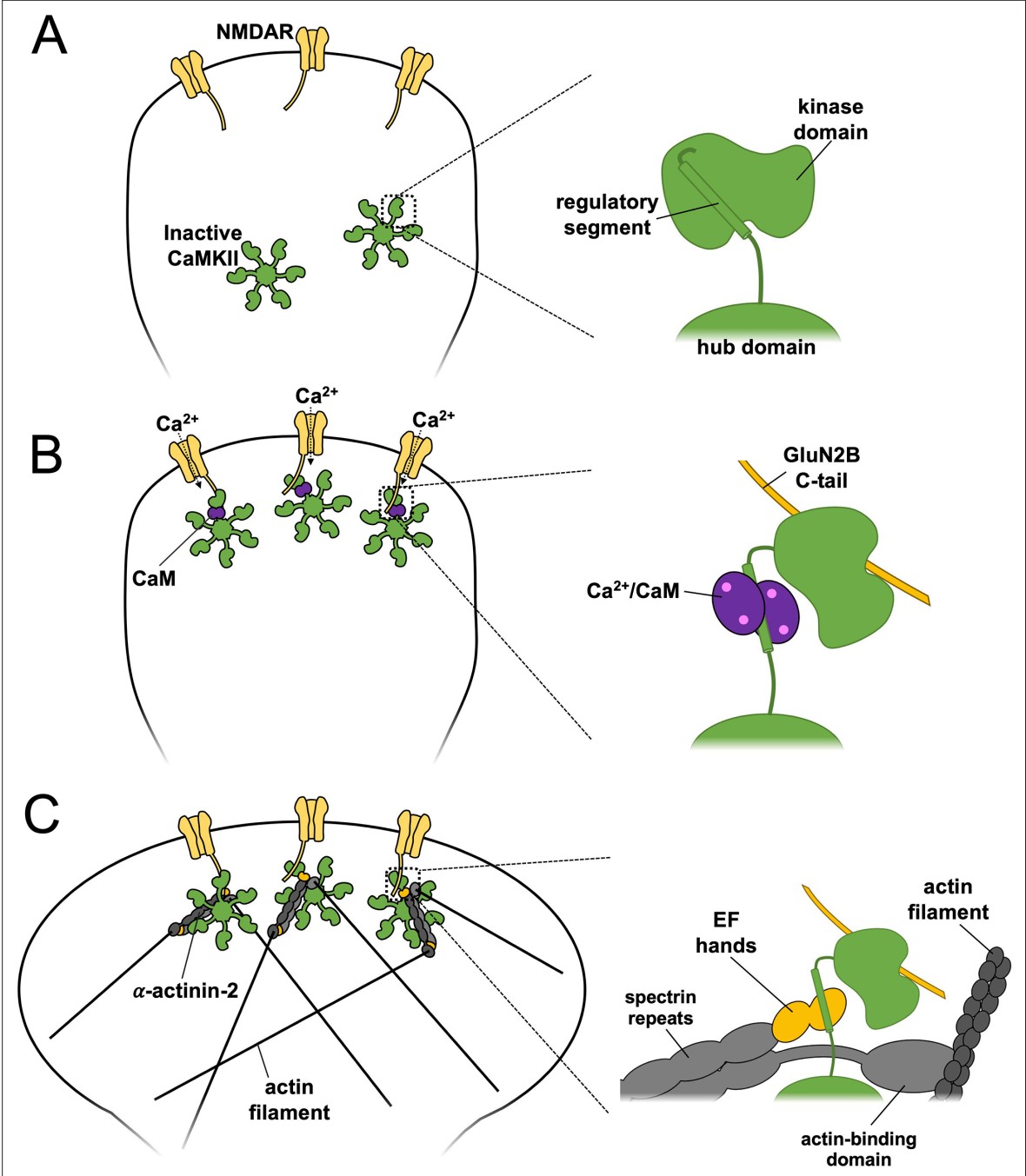

**Figure 7.** Model of actinin-calmodulin-dependent protein kinase II (CaMKII) dynamics underlying structural long-term potentiation (LTP). A three-stage model is presented with close-up illustrations of protein interactions involving CaMKII on the right for each stage. (**A**) In naïve synapses prior to $Ca^{2+}$ entry, the CaMKII regulatory segment associates with the kinase domain and is largely inaccessible to α-actinin-2. (**B**) $Ca^{2+}$ influx through NMDA receptors (NMDARs) triggers binding of $Ca^{2+}$/CaM (purple) to the regulatory segment, enabling docking of the kinase domain to the C-terminal tail of GluN2B (gold) subunits. (**C**) Following return of spine $[Ca^{2+}]$ to baseline levels, $Ca^{2+}$/CaM dissociates whereas CaMKII-NMDAR interactions persist. This combination enables α-actinin-2 to dock to the kinase domain, thereby linking the kinase to the actin cytoskeleton in support of spine enlargement.

CaMKII associated with GluN2B subunits. α-Actinins also associate with the N-terminal 13 amino acids of PSD-95, which underlie their role in spine formation (*Matt et al., 2018*), further supporting a role for the actin-crosslinking protein in the PSD. α-Actinin-2 has previously been shown to enhance binding of CaMKII to GluN2B (*Jalan-Sakrikar et al., 2012*), therefore it is logical that the reverse should also be true according to the principle of reciprocity. Our ITC measurements indicate that EF1–4 binds

to the CaMKII regulatory segment with somewhat higher affinity than EF3–4 (*Figure 3*), so there is some remaining uncertainty concerning the exact contribution of the EF1–2 region to interaction with CaMKII in neurons. Accurately conceptualising interactions between proteins in the PSD is complicated by the potential for cooperative interactions since many of the relevant proteins are oligomeric. For example, it is not clear whether EF3–4 domains at either end of an α-actinin-2 dimer could simultaneously bind to two regulatory segments within the same CaMKII dodecamer (*Penny and Gold, 2018*). Furthermore, NMDAR receptors themselves are spaced at regular intervals of ~30 nm in the PSD (*Chen et al., 2008*) – a similar scale to the width of CaMKII dodecamers (~25 nm) and the length of α-actinin-2 dimers (~28 nm) (*de Ribeiro et al., 2014*; *Myers et al., 2017*) – raising the possibility of cooperative interactions spanning multiple receptors. Developments in techniques for in situ imaging including cryo-electron tomography (*van den Hoek et al., 2022*) will be required to resolve the extent to which cooperative interactions support the structure of potentiated dendritic spines.

Our imaging experiments fit with previous reports that α-actinin-2 knockdown prevents formation of mushroom-type dendritic spines (*Hodges et al., 2014*). α-Actinin-2 is best known for its role at the Z-dics in cardiac, striated, and smooth muscle cells where it organises the lattice structure of the contractile apparatus through crosslinking actin filaments and binding titin through the EF3–4 region (*Sjöblom et al., 2008*). Actin is not visible in equivalent arrays with regular spacing in dendritic spines (*Burette et al., 2012*). In this location, α-actinins likely operate through a mechanism more analogous to their role during cytokinesis where they accumulate actin filaments at the cleavage furrow without generating a regular lattice structure (*Mukhina et al., 2007*). An aspect of CaMKII regulation of spine architecture that we did not consider in this study is the role of CaMKIIβ subunits, that bind directly to the actin cytoskeleton through elements within their regulatory segment (*O'Leary et al., 2006*; *Shen et al., 1998*). Only a fraction of CaMKII dodecamers contain β subunits (*Miller and Kennedy, 1985*) but within these assemblies it is possible that tripartite complexes assemble including CaMKII, α-actinin, and F-actin. In our summary model (*Figure 7*), we suggest that docking to NMDARs is key for enabling α-actinin-2 to access the CaMKII regulatory segment but other postsynaptic proteins that stably associate with CaMKII by occupying the substrate-binding groove of the kinase domain such as Tiam1 (*Saneyoshi et al., 2019*) could also support activity-dependent increases in CaMKII-actinin association. There is potential for therapeutic inhibition of CaMKII (*Pellicena and Schulman, 2014*), and several leading peptide inhibitors of the kinase constitute pseudosubstrate sequences that occupy the substrate-binding groove (*Reyes Gaido et al., 2023*). Our work suggests that developers of such inhibitors should be mindful of the potential for unexpected gain-of-function effects in inhibitors that increase access to the regulatory segment. Overall, our study illuminates the remarkable sophistication of regulatory processes that enable a single kinase – CaMKII – to play such a central role in the regulation of synaptic strength.

## Materials and methods
### Protein expression and purification
The EF3–4 (827–894) region of human α-actinin-2 was cloned into a modified pET28 vector using primers EF34_F & _R for expression with a Tev-cleavable N-terminal 6His-GST tag. The construct was transformed into Rosetta plysS *Escherichia coli*, and expression was induced with 1 mM IPTG after the cells had reached an $OD_{600nm}$ of ~0.7 in LB. Cells were harvested after overnight incubation at 20°C, resuspended in Ni-NTA buffer A (25 mM Tris pH 8, 500 mM NaCl, 1 mM benzamidine, 30 mM imidazole) supplemented with 1 cOmplete protease inhibitor tablet/100 mL and 0.1 mg/mL lysozyme, then clarified by centrifugation at 40,000 × *g*. The clarified lysate was incubated with Ni-NTA agarose beads (QIAGEN) for 1.5 hr. Following incubation, beads were washed and eluted (500 mM NaCl, 25 mM Tris pH 7.5, 1 mM benzamidine, 300 mM imidazole). The resulting eluate was buffer exchanged into Glutathione Sepharose binding buffer (25 mM HEPES pH 7.5, 500 mM NaCl, 1 mM DTT, 0.5 mM EDTA) and incubated with Glutathione Sepharose 4B beads for 3 hr, washed and cleaved overnight at 4°C with Tev protease. Following overnight cleavage, samples were further purified by size-exclusion chromatography with a HiLoad 16/600 Superdex 75 column equilibrated in 20 mM Na HEPES pH 7.5, 0.15 M NaCl, and 1 mM DTT.

Coding sequences for rat GluN2B (1260–1492) and α-actinin-2 EF1–4 (747–894) were cloned into pGEX-6P1 using primers BamHI_GluN2B_1260/GluN2B_Term_EcoRI and EcoRI_actinin_747/

actinin_Term_NotI, respectively. GluN2Bc was expressed in Tuner (DE3) *E. coli* whereas EF1–4 was expressed in Rosetta plysS *E. coli*. In both cases, bacteria were harvested following overnight growth in auto-inducing media (AIM) (*Studier, 2005*) at 37°C. The two protein fragments were purified in the same way. Bacteria were first lysed by sonication in lysis buffer (25 mM HEPES pH 7.5, 500 mM NaCl, 1 mM benzamidine, 1 mM DTT, 1 mM EDTA) supplemented with 1 cOmplete protease inhibitor tablet/100 mL and 0.1 mg/mL lysozyme, then clarified by centrifugation at 40,000 × *g*. Lysates were incubated with 3 mL Glutathione Sepharose 4B for 3 hr, washed and cleaved from immobilised GST fusion tag by overnight incubation with PreScission protease at 4°C. Finally, the proteins were subjected to size-exclusion chromatography with a HiLoad 16/600 Superdex 75 column equilibrated in 25 mM HEPES pH 7.5 and 150 mM NaCl. Template DNA for GluN2B (vector pCI-EGFP-NR2b) was provided by Andres Barria & Robert Malinow (RRID:Addgene_45447) (*Barria and Malinow, 2002*), and α-actinin-2 coding sequence by Kristina Djinović-Carugo (*de Ribeiro et al., 2014*).

N-terminally 6His-tagged mouse CaMKIIα was cloned into pcDNA3.1 using primers EcoRI_6His-CaMKIIa and CaMKIIa_Term_XhoI for expression in adherent HEK293T cells. Cells were transfected at 60–70% confluency with 10 µg DNA and 60 µg polyethlyenimine (MW 25000) applied to each 10 cm dish (*Curtis et al., 2022*). The media was replaced with DMEM supplemented with 10% FBS the morning after transfection, and cells were harvested 3 days after transfection. Cells were lysed by Dounce homogenisation in lysis buffer (20 mM HEPES pH 8.0, 20 mM NaCl, 2 mM DTT, 1 mM EDTA, 1 cOmplete protease inhibitor tablet/100 mL) and clarified by centrifugation for 30 min at 45,000 × *g*. CaMKII was initially enriched using anion exchange with Q Sepharose Fast Flow resin (GE Healthcare), eluted in high salt buffer (20 mM HEPES pH 7.5, 500 mM NaCl, 2 mM DTT, and 1 mM EDTA) following 2 hr incubation. This eluate was exchanged into Ni-NTA buffer A (500 mM NaCl, 25 mM Tris pH 8, 1 mM benzamidine, 20 mM imidazole) using a HiPrep desalting column (Cytiva) for binding to Ni-NTA agarose. His-CaMKIIα was eluted from the Ni-NTA agarose using a gradient into Ni-NTA buffer B (500 mM NaCl, 25 mM Tris pH 7.5, 1 mM benzamidine, 300 mM imidazole), and finally exchanged into storage buffer (25 mM HEPES pH 7.5, 150 mM NaCl, 10% wt/vol glycerol). For expression of CaMKIIα 1–315, the corresponding coding sequence was cloned into pNH-TrxT (gift of Opher Gileadi) (*Savitsky et al., 2010*) for expression with a Tev-cleavable N-terminal 6His-Trx tag in bacteria using primers pNH-Trx CaMKII_1–315_F & _R. The expression vector was transformed into Rosetta (DE3) pLysS *E. coli*, which were grown in LB and induced with 0.2 mM IPTG at $OD_{600\,nm}$ ~0.5. Cells were harvested following overnight incubation at 18°C, then lysed in Ni-NTA buffer A (25 mM Tris pH 8, 500 mM NaCl, 1 mM benzamidine, 20 mM imidazole) supplemented with 0.1 mg/mL lysozyme and 1 Complete EDTA-free protease inhibitor tablet/100 mL. 6His-Trx-CaMKIIα (1-315) was initially purified by affinity to Ni-NTA agarose, eluting with a gradient into nickel buffer B. The eluate was exchanged into anion exchange buffer A (20 mM Tris pH 8.8, 20 mM NaCl, 1 mM DTT, 1 mM EDTA) using a HiPrep 26/10 desalting column to enable binding to a Q Fast Flow column (Cytiva). Protein was eluted using a gradient into anion exchange buffer B (20 mM Tris, pH 8, 500 mM NaCl, 1 mM DTT, 1 mM EDTA). Finally, size exclusion was performed using a HiLoad Superdex 75 column equilibrated in gel filtration buffer (20 mM HEPES pH 7.5, 150 mM NaCl, 1 mM DTT).

Human CaM was expressed and purified as described previously (*Patel et al., 2017*). Briefly, untagged CaM was expressed using pET28-a in *E. coli* BL21 (DE3) cells grown in AIM. CaM was initially purified by affinity to phenyl sepharose (GE Life Sciences) in the presence of 5 mM $CaCl_2$. CaM was eluted in buffer containing 1 mM EDTA then further purified using anion exchange with a Resource Q column (GE Life Sciences) before dialysis into water and lyophilisation in a vacuum concentrator.

## Crystallography

For crystallisation of α-actinin-2 EF3–4 with CaMKIIα regulatory segment, peptide corresponding to CaMKIIα positions 294–315 (sequence NARRKLKGAILTTMLATRNFSG, acetylated at N-terminus, amidated at C-terminus) was synthesised by Biomatik at >95% purity. α-Actinin-2 EF3–4 (16 mg/mL) was mixed with a 2.5-fold molar excess of CaMKIIα peptide in a precipitant solution containing 0.1 M Bis-Tris pH 6.5, 25% wt/vol polyethylene glycol 3350, and crystals were grown by sitting drop vapor diffusion at 24°C. Diffraction data was collected to high resolution at Diamond Light Source beamline I04 with X-rays at a wavelength of 0.9795 Å, and at beam line P13 at the PETRA III storage ring (DESY, Hamburg, Germany) with X-rays tuned to 2.0664 Å to amplify the anomalous signal of sulfur. Diffraction data was reduced using DIALS (*Winter et al., 2018*) and scaled with Aimless (*Evans and

*Murshudov, 2013*) before experimental phasing using single-wavelength anomalous diffraction of native sulfur atoms in CRANK2 (*Skubák et al., 2018*). Refinement was completed in PHENIX (*Liebschner et al., 2019*). The following residues were omitted from the final model since they could not be clearly resolved in the electron density: EF3–4 892–894 (chain B); CaMKIIα 314–315 (chain C); and CaMKIIα 311–315 (chain D). Full data collection and refinement statistics are provided in *Supplementary file 1*.

## Isothermal titration calorimetry

ITC experiments were performed using a MicroCal PEAQ (Malvern Panalyticial). To investigate binding between CaMKII peptides and EF1–4/EF3–4, experiments were performed at 25°C in 25 mM HEPES pH 7.5 and 150 mM NaCl, where 500 μM CaMKII peptides were injected into a cell containing 50 μM EF1–4/EF3–4 using 2 μL injections at 2 min intervals and a constant mixing speed of 750 rpm. Measurements between Trx-CaMKIIα 1–315 and EF3–4 were performed using 30 μM Trx-CaMKIIα 1–315 and 300 μM of EF hands 3–4 and additionally supplementing the buffer with 2 mM $MgCl_2$ and 1 mM ADP. To investigate CaMKII-CaM interactions, ITC was performed at 10°C using 30 μM Trx-CaMKIIα 1–315 – 300 μM CaM (buffer supplemented with 2 mM $MgCl_2$, 1 mM ADP, and 1 mM $CaCl_2$); and 30 μM CaM – 300 μM CaMKII regulatory peptide (buffer supplemented with 1 mM $CaCl_2$). Protein concentrations were determined using a combination of absorbance at $A_{280}$ (with reference to theoretical extinction coefficients) and bicinchoninic acid assays. Single protein preparations were used for experimental replicates with the exception of EF3–4, where two preparations were used. Each preparation of EF3–4 yielded similar results. All ITC measurements were collected, baselined, and integrated before non-linear least-squares fitting to single binding models using MicroCal origin software.

## Culture and transfection of primary hippocampal neurons

Primary hippocampal cultures were prepared from E18 Sprague-Dawley rats and plated onto 13 mm coverslips, pre-treated with poly-L-lysine (1 mg/mL) at a density of $1×10^5$ cells per coverslip. Neurons were cultured in neurobasal medium supplemented with B27, GlutaMAX, and Penicillin/Streptomycin. On DIV10, neurons were transfected with 0.8 μg DNA and 2 μL Lipofectamine 2000 per 13 mm coverslip in six-well plates. For double transfections, 0.4 μg of each vector was included in the transfection mixture in all cases. After transferring transfected neurons into fresh media on DIV11, in vitro culture was continued until chemical LTP/fixing on DIV14.

pIRES2-GFP vectors containing N-terminally FLAG-tagged coding sequences were constructed using primers listed in *Supplementary file 2*. The L854R mutation was introduced into pcDNA3.1-FLAG-α-actinin-2 (747–894) by site-directed mutagenesis with primers L854R_F & _R. pIRES2-EGFP vector for expressing N-terminally V5-tagged CaMKIIα was generated using primers BamHI_V5_CaMKII_F and CaMKII_Term_SalI_R. Site-directed mutagenesis was used to generate alanine substitutions in this vector as follows: T286A_F/R for the T286A variant; T305A_F/R for T305; and T306A_F/R for T306A. For expression of GFP-GluN2B subunits containing internal HA tags, we used primers AgeI_4gHA4g_F and EcoRI_GluN2B_R to insert a 4Gly-HA-4Gly sequence after position G1211 in a GFP-GluN2B vector (*Luo et al., 2002*). We used site-directed mutagenesis with primers CaMKIIa_AQ_F and CaMKIIa_AQ_R to generate the ΔCaMKII variant of this construct.

## Chemical LTP

Chemical LTP (cLTP) was induced in cultured hippocampal neurons by activating NMDARs with glycine as described previously (*Fortin et al., 2010*; *McLeod et al., 2018*). Primary hippocampal neurons were first transferred into control solution (5 mM HEPES pH 7.4, 125 mM NaCl, 2.5 mM KCl, 1 mM $MgCl_2$, 2 mM $CaCl_2$, 33 mM D-glucose, 20 μM D-APV, 3 μM strychnine, 20 μM bicuculline, 0.5 μM TTX) for 20 min at room temperature (RT). cLTP was induced by incubating for 10 min at RT in cLTP solution (5 mM HEPES pH 7.4, 125 mM NaCl, 2.5 mM KCl, 2 mM $CaCl_2$, 33 mM D-glucose, 3 μM strychnine, 20 μM bicuculline, 0.2 mM glycine). Following cLTP induction, neurons were returned to control solution for between 1 and 4 hr before fixation in fixing solution (PBS supplemented with 4% paraformaldehyde, 4% sucrose, 0.2% glutaraldehyde). For PLA assays, neurons were fixed 2 hr after cLTP induction other than in the time course experiment (*Figure 2A*), in which they were fixed either 10, 30, 120, 600, or 3600 s after the initial addition of cLTP solution. For the longest time-point, the neurons were returned to control solution after 10 min.

## PLAs and confocal imaging

PLAs were performed using reagents from Duolink In Situ PLA kits. Fixed neurons were permeabilised for 5 min at RT in PBS supplemented with 1% BSA/0.1% Triton X-100, and blocked for 1 hr in PBS supplemented with 10% BSA before overnight incubation at 4°C with primary antibodies (mouse anti-CaMKIIα, 1 in 400 dilution; goat anti-FLAG, 1 in 200 dilution; rabbit anti-GFP, 1 in 300 dilution; rabbit anti-HA, 1 in 500 dilution, mouse anti-V5, 1 in 500 dilution) in PBS supplemented with 1% BSA. On the following morning, neurons were incubated with the corresponding Duolink probes (anti-goat PLUS with either anti-mouse MINUS or anti-rabbit MINUS) and goat anti-rabbit Alexa Fluor-405 (Thermo Fisher) at 37°C for 1 hr. Probes were ligated at 37°C for 30 min and signals were amplified at 37°C for 100 min. Neurons were imaged using ZEN software and a 60× oil objective NA = 1.40 using either a Zeiss LSM 700 inverted microscope (data presented in *Figures 1 and 3*) or an LSM 780 microscope equipped with an airyscan module (data presented in *Figures 2 and 6*). For PLA experiments, images were collected using 405 nm excitation/421 nm emission for GFP, and 594 nm excitation/619 nm emission for PLA puncta. For imaging neurons transfected with α-actinin-2 EF1–4 disruptors, intrinsic GFP fluorescence was imaged using 488 nm excitation/509 nm emission. Data were obtained from at least three independent neuronal cultures unless otherwise stated. Images were analysed using Neuron-Studio software (Icahn School of Medicine at Mount Sinai) to determine spine width and morphology; and the Distance Analysis (DiAna) plugin (*Gilles et al., 2017*) for ImageJ (NIH) to identify PLA puncta.

## Magnetic bead pull-down assays

For each CaMKIIα pull-down assay, 0.25 µg of Pierce Glutathione Magnetic Agarose Beads were charged with 4 µg GST or GST-EF1–4 in basic binding buffer (25 mM HEPES pH 7.5, 150 mM NaCl, 0.05% Tween-20, 10 mM $MgCl_2$, 0.5 mM ADP, 2 mM DTT) for 2 hr at 4°C. Protein mixtures containing CaMKII, CaM, and GluN2Bc (as appropriate) were separately pre-incubated for 1 hr in basic binding buffer supplemented with $CaCl_2$. These mixtures were diluted with equal volumes of basic binding buffer containing 4% BSA to achieve final concentrations of 2 mM $CaCl_2$, 2% BSA, 0.1 µM CaMKIIα, 3 µM CaM, and 1.5 µM GluN2Bc as appropriate. For each assay, 100 µL protein mixture was incubated with GST/GST-EF1–4 magnetic beads for 20 min before the reactions were supplemented with either 10, 2.5, or 1.8 mM EGTA. Free $Ca^{2+}$ concentrations were estimated using maxchelator (*Bers et al., 2010*). Following 20 min further incubation, each pull-down was washed with 3×500 µL basic binding buffer supplemented with 2 mM $CaCl_2$ and the appropriate concentration of EGTA. Proteins were eluted from the beads by incubation with 50 µL 1× LDS loading buffer (5 min heating at 65°C). CaMKII and GST/GST-EF1–4 were detected using immunoblotting with mouse anti-CaMKIIα and rabbit anti-GST antibodies.

## Statistical analysis

Data were assessed for normality using Kolmogorov-Smirnov testing. Normally distributed data were analyzed using unpaired two-tailed Student's t-tests whereas Mann-Whitney tests were applied for non-parametric data. *$p < 0.05$; **$p < 0.01$; ***$p < 0.001$. Two-way ANOVA was utilised to analyse PLA measurements with neurons expressing GluN2B-iHA. Non-linear curve fitting by iterative least squares minimisation was performed in Origin (OriginLab) to fit PLA time course data to a Hill function.

# Acknowledgements

This work was supported by BBSRC project grant BB/N015274/1, a Wellcome Trust and Royal Society Sir Henry Dale fellowship to MGG (104194/Z/14/Z), and a BBSRC studentship to AC. We are grateful to Opher Gileadi, Andres Barria, Robert Malinow, and Kristina Djinović-Carugo for providing DNA vectors in support of this work.

## Additional information

### Funding

| Funder | Grant reference number | Author |
| --- | --- | --- |
| Wellcome Trust | 104194/Z/14/Z | Matthew G Gold |
| Biotechnology and Biological Sciences Research Council | BB/N015274/1 | Jian Zhu Christopher J Penny Matthew G Gold |
| Biotechnology and Biological Sciences Research Council | 2081382 | Ashton J Curtis |

The funders had no role in study design, data collection and interpretation, or the decision to submit the work for publication. For the purpose of Open Access, the authors have applied a CC BY public copyright license to any Author Accepted Manuscript version arising from this submission.

### Author contributions

Ashton J Curtis, Investigation, Writing - original draft, Writing - review and editing; Jian Zhu, Investigation, Writing - review and editing; Christopher J Penny, Investigation; Matthew G Gold, Conceptualization, Supervision, Funding acquisition, Investigation, Writing - original draft, Writing - review and editing

### Author ORCIDs

Ashton J Curtis http://orcid.org/0009-0008-8320-0169
Matthew G Gold http://orcid.org/0000-0002-1281-0815

### Ethics

EthicsExperiments involving rats were performed in accordance with the United Kingdom Animals Act, 1986 and within University College London Animal Research guidelines overseen by the UCL Animal Welfare and Ethical Review Body under project code 14058.

### Decision letter and Author response

Decision letter https://doi.org/10.7554/eLife.85008.sa1
Author response https://doi.org/10.7554/eLife.85008.sa2

## Additional files

### Supplementary files

• Supplementary file 1. Data collection and refinement statistics.
• Supplementary file 2. Oligonucleotide primers sequences. Primer sequences are shown (5' to 3').
• MDAR checklist

### Data availability

Coordinates and structure factors have been deposited with the RCSB Protein Databank for the EF3-4 - CaMKII regulatory segment peptide complex with accession ID 6TS3.

The following dataset was generated:

| Author(s) | Year | Dataset title | Dataset URL | Database and Identifier |
| --- | --- | --- | --- | --- |
| Zhu J, Gold M G | 2021 | EF-hands 3 and 4 of alpha-actinin in complex with CaMKII regulatory segment | https://www.rcsb.org/structure/6ts3 | RCSB Protein Data Bank, 6TS3 |

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

# Appendix 1

### Appendix 1—key resources table

| Reagent type (species) or resource | Designation | Source or reference | Identifiers | Additional information |
|---|---|---|---|---|
| Strain, strain background (*Escherichia coli*) | TOP10 chemically competent | Life Technologies | Cat# C404003 | |
| Strain, strain background (*Escherichia coli*) | BL21 (DE3) | Thermo Fisher Scientific | Cat# EC0114 | |
| Strain, strain background (*Escherichia coli*) | BL21 Tuner (DE3) pLysS | Merck | Cat# 70624 | |
| Strain, strain background (*Escherichia coli*) | BL21 Star (DE3) | Thermo Fisher Scientific | Cat# C601003 | |
| Cell line (*Homo sapiens*) | HEK293T | ATCC | Cat# CRL-3216 | Verified using STR profiling, *Mycoplasma*-tested |
| Biological sample (*Rattus norvegicus*) | Sprague-Dawley rat | UCL breeding colony | Not applicable | |
| Antibody | Rabbit polyclonal anti-GFP | Sigma-Aldrich | Cat# SAB4301138; RRID:AB_2750576 | 10 µg/mL |
| Antibody | Goat polyclonal anti-FLAG | Novus | Cat# NB600-344; RRID:AB_10000565 | 5 µg/mL |
| Antibody | Rabbit polyclonal anti-GST | Merck | Cat# G7781; RRID:AB_259965 | 0.2 µg/mL |
| Antibody | Mouse monoclonal anti-CaMKIIα | Santa Cruz Biotechnology | Cat# sc-32288; RRID:AB_626787 | 0.2 µg/mL for IB; 0.5 µg/mL for ICC |
| Antibody | Rabbit polyclonal anti-HA | Abcam | Cat# ab9110; RRID:AB_307019 | 2 µg/mL |
| Antibody | Mouse monoclonal anti-V5 | Invitrogen | Cat# MA5-15253; RRID: AB_10977225 | 2 µg/mL |
| Antibody | Goat polyclonal anti-rabbit HRP-linked secondary antibody | Cell Signaling Technology | Cat# 7074 S; RRID:AB_2099233 | 0.5 µg/mL |
| Antibody | Goat polyclonal anti-mouse HRP secondary antibody | Thermo Fisher Scientific | Cat# 32230; RRID:AB_1965958 | 0.5 µg/mL |
| Antibody | Goat polyclonal anti-rabbit Alexa Fluor-405 | Thermo Fisher Scientific | Cat# A-31556; RRID:AB_221605 | 4 µg/mL |
| Antibody | Duolink In Situ PLA Probe, Donkey Polyclonal Anti-Goat PLUS | Merck | Cat# DUO92003 | At 1x final concentration according to the manufacturer's guidelines. |
| Antibody | Duolink In Situ PLA Probe, Donkey Polyclonal Anti-Mouse MINUS | Merck | Cat# DUO92004 | At 1x final concentration according to the manufacturer's guidelines. |
| Antibody | Duolink In Situ PLA Probe, Donkey Polyclonal Anti-Rabbit MINUS | Merck | Cat# DUO92005 | At 1x final concentration according to the manufacturer's guidelines. |
| Antibody | Rabbit anti-GFP | Sigma-Aldrich | Cat# SAB4301138; RRID:AB_2750576 | 10 µg/mL |
| Antibody | Goat anti-FLAG | Novus | Cat# NB600-344; RRID:AB_10000565 | 5 µg/mL |
| Antibody | Rabbit anti-GST | Merck | Cat# G7781; RRID:AB_259965 | 0.2 µg/mL |
| Antibody | Mouse anti-CaMKIIα | Santa Cruz Biotechnology | Cat# sc-32288; RRID:AB_626787 | 0.2 µg/mL for IB; 0.5 µg/mL for ICC |
| Antibody | Rabbit anti-HA | Abcam | Cat# ab9110; RRID:AB_307019 | 2 µg/mL |

*Appendix 1 Continued on next page*

*Appendix 1 Continued*

| Reagent type (species) or resource | Designation | Source or reference | Identifiers | Additional information |
|---|---|---|---|---|
| Antibody | Mouse anti-V5 | Invitrogen | Cat# MA5-15253; RRID:AB_10977225 | 2 µg/mL |
| Antibody | Goat anti-rabbit HRP-linked secondary antibody | Cell Signaling Technology | Cat# 7074 S; RRID:AB_2099233 | 0.5 µg/mL |
| Antibody | Goat anti-mouse poly-HRP secondary antibody | Thermo Fisher Scientific | Cat# 32230; RRID:AB_1965958 | 0.5 µg/mL |
| Antibody | Goat anti-rabbit Alexa Fluor-405 | Thermo Fisher Scientific | Cat# A-31556; RRID:AB_221605 | 4 µg/mL |
| Recombinant DNA reagent | pIRES2-EGFP | Clontech | Cat# 6029-1 | |
| Recombinant DNA reagent | pIRES2-EGFP-FLAG-α-actinin-2 construct series (1–894; 1–748; 747–894; 747–890; 1–890; 747–894 L854R) | This study | Not applicable | Contact MGG to obtain this vector |
| Recombinant DNA reagent | pcDNA3.1-6His-CaMKIIα | This study | Not applicable | Contact MGG to obtain this vector |
| Recombinant DNA reagent | pIRES2-EGFP-V5-CaMKIIα (WT, T286A, T305A, and T306A variants) | This study | Not applicable | Contact MGG to obtain this vector |
| Recombinant DNA reagent | pET28-6His-GST-Tev- actinin-2 EF3–4 (827–894) | This study | Not applicable | Contact MGG to obtain this vector |
| Recombinant DNA reagent | pNH-TrxT-6His-Trx-Tev-CaMKIIα (1–315) | This study | Not applicable | Contact MGG to obtain this vector |
| Recombinant DNA reagent | pGEX6P1-α-actinin-2 EF1–4 (747–894) | This study | Not applicable | Contact MGG to obtain this vector |
| Recombinant DNA reagent | pGEX6P1-GluN2B (1260–1492) | This study | Not applicable | Contact MGG to obtain this vector |
| Recombinant DNA reagent | pET28-CaM (untagged) | *Patel et al., 2017* | Not applicable | Contact MGG to obtain this vector |
| Recombinant DNA reagent | pCI-EGFP-NR2B | Andres Barria and Robert Malinow/Addgene | RRID:Addgene_45447 | |
| Recombinant DNA reagent | pEGFP-NR2B vector | Stefano Vicini/Addgene | RRID:Addgene_17925 | |
| Recombinant DNA reagent | pEGFP-NR2B-iHA (WT and ΔCaMKII variants) | This study | Not applicable | Contact MGG to obtain this vector |
| Recombinant DNA reagent | pNH-TrxT | Opher Gileadi/Addgene | RRID:Addgene_26106 | |
| Recombinant DNA reagent | pET3d-6His-α-actinin-2 | Kristina Djinović-Carugo (*de Ribeiro et al., 2014*) | Not applicable | |
| Chemical compound, drug | Lipofectamine 2000 | Thermo Fisher Scientific | Cat# 11668019 | |
| Chemical compound, drug | Polyethylenimine (linear, MW25000) | Polysciences | Cat# 23966 | |
| Chemical compound, drug | Duolink In Situ Detection Reagents Red | Merck | Cat# DUO92008 | |
| Chemical compound, drug | Pierce Glutathione Magnetic Agarose Beads | Thermo Fisher Scientific | Cat# 78601 | |
| Chemical compound, drug | DMEM, high glucose, pyruvate | Thermo Fisher Scientific | Cat# 41966029 | |
| Chemical compound, drug | Trypsin | Thermo Fisher Scientific | Cat# 25300054 | |
| Chemical compound, drug | Penicillin/Streptomycin | Thermo Fisher Scientific | Cat# 15140122 | |
| Chemical compound, drug | GlutaMAX | Thermo Fisher Scientific | Cat# 35050061 | |
| Chemical compound, drug | DPBS, no calcium, no magnesium | Thermo Fisher Scientific | Cat# 14190144 | |

*Appendix 1 Continued on next page*

*Appendix 1 Continued*

| Reagent type (species) or resource | Designation | Source or reference | Identifiers | Additional information |
|---|---|---|---|---|
| Chemical compound, drug | Neurobasal-A medium | Thermo Fisher Scientific | Cat# 10888022 | |
| Chemical compound, drug | B27 supplement | Gibco | Cat# 17504044 | |
| Chemical compound, drug | Poly-L-Lysine | Sigma-Aldrich | Cat# P2636 | |
| Chemical compound, drug | Complete, Mini, EDTA-free Protease Inhibitor Cocktail | Roche | Cat# 11836170001 | |
| Software, algorithm | ImageJ (version 1.52) | NIH | RRID:SCR_003070 | |
| Software, algorithm | NeuronStudio | *Rodriguez et al., 2008* | https://icahn.mssm.edu; RRID:SCR_013798 | |
| Software, algorithm | Origin | OriginLab | RRID:SCR_014212 | |

