## [Editor Report]

This fundamental study substantially advances our understanding of the synaptic targeting of a major postsynaptic protein kinase, CaMKII, that underlies the persistence of excitatory synaptic strength in synaptic plasticity. The evidence for the authors' claims is compelling, with cell biological, biochemical, as well as structural biological approaches. This work will be of broad interest to cell and computational biologists working on learning/memory.

---

## [Decision Letter]

**Decision letter after peer review:**

Thank you for submitting your article "Molecular basis of interactions between CaMKII and α-actinin-2 that underlie dendritic spine enlargement" for consideration by *eLife*. Your article has been reviewed by 3 peer reviewers, one of whom is a member of our Board of Reviewing Editors, and the evaluation has been overseen by Volker Dötsch as the Senior Editor. The following individual involved in the review of your submission has agreed to reveal their identity: Johannes W Hell (Reviewer #2).

Essential revisions:

As you will see in the comments below, the reviewers find the model that synaptic localization of CaMKII and α-actinin-2 complex during structural LTP that is cooperated by interaction with the NMDAR subunit GluN2B is of broad interest. As it stands, all three reviewers agree that the results are convincing but would require the inclusion of additional studies and clarification of the model, in particular with respect to its co-localization of GluN2B as well as the roles of autophosphorylation of CaMKII, before they could recommend publication in *eLife*.

All reviewers reached a consensus to ask the authors for revision as follows:

1) the PLA assay between α-actinin-2 and either CaMKII (as already partially done) or GluN2B should be done.

2) The effects of autophosphorylation sites (T286, T305, T306) of CaMKII have to be tested individually in the neuron.

3) Time course study for the puncta formation of the complex, such as CaMKII/GluN2B, CaMKII/ α-actinin-2, or GluN2B/ α-actinin-2. It does not have to show the three proteins together.

4) Regarding the roles for EF1 and 2 in α-actinin-2 co-localization with CaMKII in the neuron (Item2 of reviewer 3), the interpretation of the role of EF1/2 in the absence of any new supportive data would be overestimating. The contribution of EF1/2 can be addressed by the PLA assay with α-actinin-2 mutants such as δ-EF1/2, δ-EF3/4, or δ-EF1-4 in Figure 1. Alternatively, just soften the interpretations of the existing data.

*Reviewer #1 (Recommendations for the authors):*

1. Why did the authors use proximity ligation assays (PLAs) to detect the interaction between CaMKII and α-actinin-2? Is this method able to compare the amount of GluN2B and α-actinin-2 in the complex with CaMKII at synapse during LTP? If so, it will be useful to the author's proposal.

2. Are GluN2B binding to CaMKII or phosphorylation on T286 of CaMKII associated with affinity to α-actinin-2?

*Reviewer #2 (Recommendations for the authors):*

1. The authors equate an increase in PLA signal as an increase in aACTN binding to CaMKII; however, PLA is only a measure of proximity but not necessarily of a specific interaction. This aspect needs to be clearly spelled out and taken into account in the interpretations of the results.

*Reviewer #3 (Recommendations for the authors):*

Additional studies could be performed to provide more substantial support for the model presented.

1. A central part of the final model is that GluN2B stabilizes the CaMKII-actinin complex. However, the manuscript contains no data to show that CaMKII-binding to GluN2B is important for CaMKII-actinin colocalization, or even to show that GluN2B colocalizes with the CaMKII-actinin complexes.. The impact of the study would be significantly enhanced by additional neuronal studies to demonstrate the importance of GluN2B in CaMKII-actinin colocalization.

2. The current in vitro and structural data confirm the critical role of actinin EF3 and EF4 domains in binding to the CaMKII regulatory domain, and further indicate that the actinin EF1 and EF2 domains can increase the affinity ~3-fold. This seems like a rather modest effect and the interpretation is critically dependent on the precise determination of the concentrations of the various proteins. However, the methods do not seem to specify how protein concentrations were determined and do not indicate whether experimental replicates were conducted with the same protein preparation or independently prepared preparations. Furthermore, there is no evidence that EF1/2 domains specifically play a role in CaMKII-actinin colocalization in neurons and dendritic spine enlargement. The impact of the study would be significantly enhanced by additional studies testing the role of EF1 and EF2 in neurons.

3. Prior studies have found that CaMKII autophosphorylation at Thr286, Thr305 and Thr306 dynamically modulate CaMKII targeting to the synapse. Specifically: 1. Autophosphorylation at Thr286 modulates CaMKII binding to GluN2B, apparently with little direct effect on binding to actinin; but, there may be an indirect effect on CaMKII-actinin colocalization mediated by changes in GluN2B binding. 2. Autophosphorylation at Thr305 OR Thr306 can interfere with CaM binding, but only Thr306 autophosphorylation interferes with actinin binding. However, the mechanism underlying the disruption of postsynaptic targeting by autophosphorylation at Thr305 and/or Thr306 is unclear. The final model predicts that autophosphorylation at each of these sites may modulate CaMKII-actinin interaction. The impact of the study would be significantly enhanced by additional studies using T286A, T305A, and T306A mutant CaMKIIs to determine the role of autophosphorylation at these sites on actinin-CaMKII colocalization and the spine size.

4. There is a concern about the apparent use of t-tests in all studies, even when multiple comparisons are being made in some datasets. Wouldn't ANOVAs be more appropriate?

---

## [Author Response]

Essential revisions:As you will see in the comments below, the reviewers find the model that synaptic localization of CaMKII and α-actinin-2 complex during structural LTP that is cooperated by interaction with the NMDAR subunit GluN2B is of broad interest. As it stands, all three reviewers agree that the results are convincing but would require the inclusion of additional studies and clarification of the model, in particular with respect to its co-localization of GluN2B as well as the roles of autophosphorylation of CaMKII, before they could recommend publication in eLife.All reviewers reached a consensus to ask the authors for revision as follows:1) the PLA assay between α-actinin-2 and either CaMKII (as already partially done) or GluN2B should be done.2) The effects of autophosphorylation sites (T286, T305, T306) of CaMKII have to be tested individually in the neuron.3) Time course study for the puncta formation of the complex, such as CaMKII/GluN2B, CaMKII/ α-actinin-2, or GluN2B/ α-actinin-2. It does not have to show the three proteins together.4) Regarding the roles for EF1 and 2 in α-actinin-2 co-localization with CaMKII in the neuron (Item2 of reviewer 3), the interpretation of the role of EF1/2 in the absence of any new supportive data would be overestimating. The contribution of EF1/2 can be addressed by the PLA assay with α-actinin-2 mutants such as δ-EF1/2, δ-EF3/4, or δ-EF1-4 in Figure 1. Alternatively, just soften the interpretations of the existing data.

We have performed additional experiments as requested. Regarding each of these four points: (1) We have performed new PLA experiments to monitor interactions between GluN2B subunits and either CaMKII (Figure 6G and Figure 6—figure supplement 2) or α-actinin-2 (Figure 6H and Figure 6—figure supplement 3). We have also undertaken further analysis of our existing PLA data for interactions between a-actinin-2 and CaMKII to establish the frequency of puncta formation by spine type before and after NMDAR activation (Figure 1—figure supplement 2). (2) We have completed PLA experiments comparing interaction of V5-CaMKIIα WT/T286A/T305A/T306A and FLAG-a-actinin-2 (Figure 2B and C, Figure 2—figure supplement 2). (3) We now include time-course PLA measurements to establish how quickly CaMKIIα and α-actinin-2 associate following NDMAR activation (Figure 2A and Figure 2—figure supplement 1). (4) We have softened the interpretation of our existing data regarding the contribution of EF hands 1-2 (lines 457-459). These additions to our study are described in detail in the point-by-point responses to each reviewer.

Reviewer #1 (Recommendations for the authors):1. Why did the authors use proximity ligation assays (PLAs) to detect the interaction between CaMKII and α-actinin-2? Is this method able to compare the amount of GluN2B and α-actinin-2 in the complex with CaMKII at synapse during LTP? If so, it will be useful to the author's proposal.

As recommended by the editor, and in line with the reviewer’s comment, we have performed additional PLA experiments to investigate interactions between CaMKII and GluN2B subunits before and after NMDAR activation. The results of these experiments are shown in Figure 6 panels F-H, and in Figure 6—figure supplements 2 and 3. We generated a vector for expression of GFP-GluN2B bearing an internal HA tag near to the CaMKII binding site (Figure 6F) that enabled us to monitor interactions between the receptor subunit and either CaMKII (Figure 6G) and a-actinin-2 (Figure 6H). We found that in both cases interactions were elevated following NMDAR activation, and that mutation of the CaMKII docking site in GluN2B suppressed the extent to which interactions with both α-actinin-2 and CaMKII were elevated following NMDAR activation. In addition to the new figures, we have included text related to these experiments in the results (lines 389-419) and methods (lines 602-605) sections, and figure legends for the new figures are provided on lines 1063-1070, 1167-1172, and 11741180.

2. Are GluN2B binding to CaMKII or phosphorylation on T286 of CaMKII associated with affinity to α-actinin-2?

In our revised submission, we include new experiments to investigate whether T286 phosphorylation is required for interactions between CaMKII and α-actinin-2. The results of these experiments are shown in Figure 2B and C and Figure 2—figure supplement 2. We used PLA to monitor interactions between V5-CaMKIIa variants and FLAG-α-actinin-2. Incorporation of the phospho-null mutation T286A in CaMKIIα did not alter PLA puncta formation relative to WT CaMKIIα in either neurons before or after NMDAR activation. These results are discussed in full on lines 151-177. We have also updated the methods section on lines 598-601 and 627, and new figure legends are provided on lines 1014-1020, and 1109-11106. New experiments investigating the role of GluN2B binding to CaMKII are described in Reviewer 1, point1.

Reviewer #2 (Recommendations for the authors):1. The authors equate an increase in PLA signal as an increase in aACTN binding to CaMKII; however, PLA is only a measure of proximity but not necessarily of a specific interaction. This aspect needs to be clearly spelled out and taken into account in the interpretations of the results.

The reviewer raises an important point that positive PLA signal does not necessarily indicate direct protein-protein interaction since rolling circle amplification can occur when the target protein pair is separated up to a maximum theoretical distance of 40 nm (Trifilieff et al., Biotechniques, 2011, PMID 21806555; https://www.sigmaaldrich.com/GB/en/products/protein-biology/duolinkproximity-ligation-assay). In the case of experiments focused on interactions between α-actinin-2 and CaMKII, we included controls including imaging with α-actinin-2 constructs lacking the EF1-4 region (Figure 1D), and an EF1-4 construct bearing mutations to prevent association with CaMKII (Figure 2—figure supplement 1C). These experiments indicate that the PLA puncta we observe are likely to correspond to direct interactions between α-actinin-2 and CaMKII. We have included the following additional text on lines 123-125 to explain how the limitations of PLA should be factored into the interpretation of our results: “Although PLA signal is a measure of protein proximity rather than direct protein-protein interaction, the low levels of puncta formation observed with the ∆EF1-4 construct indicate that puncta formation in this case is likely to correspond to direct interaction between CaMKII and α-actinin-2”.

Reviewer #3 (Recommendations for the authors):Additional studies could be performed to provide more substantial support for the model presented.1. A central part of the final model is that GluN2B stabilizes the CaMKII-actinin complex. However, the manuscript contains no data to show that CaMKII-binding to GluN2B is important for CaMKII-actinin colocalization, or even to show that GluN2B colocalizes with the CaMKII-actinin complexes.. The impact of the study would be significantly enhanced by additional neuronal studies to demonstrate the importance of GluN2B in CaMKII-actinin colocalization.

To investigate the role of GluN2B interactions with CaMKII and a-actinin-2, we have generated a new construct – GFP-GluN2B-iHA – that includes an internal HA tag to enable in situ PLA assays of interactions to the GluN2B C-terminal tail (Figure 6F). Our revised submission includes two sets of experiments using WT and ∆CaMKII (L1298A/R1300Q) variants of this construct to monitor interactions with CaMKII (Figure 6G and Figure 6—figure supplement 2) and α-actinin-2 (Figure 6H and Figure 6—figure supplement 3). We found that PLA puncta formation indicative of GluN2B-CaMKII interaction was elevated as expected following NMDAR activation, and that this effect was attenuated by mutation of the CaMKII anchoring site. We also detected interactions between GluN2B and α-actinin-2 that were elevated (to a lesser extent) upon NMDAR activation. Mutation of the GluN2B CaMKII anchoring site also suppressed elevation of GluN2B-α-actinin-2 interactions following NMDAR activation. We performed two-way ANOVAs to show that there was interaction between mutation of the CaMKII anchoring site and NMDAR activation for both GluN2B-CaMKII and GluN2B-α-actinin-2 interactions. In this way, these additional experiments are consistent with the notion that CaMKII docking to GluN2B plays at least some role in increasing access to α-actinin-2 in structural LTP. These new experiments are described in full on lines 389-419. We also include additional methodological information related to these experiments on lines 602-605, and include new figure legends on lines 10631070, 1167-1172, and 1174-1180.

2. The current in vitro and structural data confirm the critical role of actinin EF3 and EF4 domains in binding to the CaMKII regulatory domain, and further indicate that the actinin EF1 and EF2 domains can increase the affinity ~3-fold. This seems like a rather modest effect and the interpretation is critically dependent on the precise determination of the concentrations of the various proteins. However, the methods do not seem to specify how protein concentrations were determined and do not indicate whether experimental replicates were conducted with the same protein preparation or independently prepared preparations. Furthermore, there is no evidence that EF1/2 domains specifically play a role in CaMKII-actinin colocalization in neurons and dendritic spine enlargement. The impact of the study would be significantly enhanced by additional studies testing the role of EF1 and EF2 in neurons.

To clarify our experimental approach for ITC experiments, we have included additional information in the methods section on lines 574-577, as follows: “Protein concentrations were determined using a combination of absorbance at A_280_ (with reference to theoretical extinction coefficients) and bichinchoninic acid (BCA) assays. Single protein preparations were used for experimental replicates with the exception of EF3-4, where two preparations were used. Each preparation of EF3-4 yielded similar results”. We agree with the reviewer that the effect of including the EF1-2 region on the affinity is modest, and whether the EF1-2 hands contribute to binding the regulatory segment does not fundamentally alter the mechanism put forward in Figure 6. Therefore, we decided to prioritise timecourse experiments, and experiments involving GluN2B and CaMKII phosphorylation site mutants. In the discussion, we have included the following additional text (lines 457-459) to highlight that there is some uncertainty regarding the contribution of the EF1-2 hands: “Our ITC measurements indicate that EF1-4 binds to the CaMKII regulatory segment with somewhat higher affinity than EF3-4 (Figure 3), so there is some remaining uncertainty concerning the exact contribution of the EF1-2 region to interaction with CaMKII in neurons”.

3. Prior studies have found that CaMKII autophosphorylation at Thr286, Thr305 and Thr306 dynamically modulate CaMKII targeting to the synapse. Specifically: 1. Autophosphorylation at Thr286 modulates CaMKII binding to GluN2B, apparently with little direct effect on binding to actinin; but, there may be an indirect effect on CaMKII-actinin colocalization mediated by changes in GluN2B binding. 2. Autophosphorylation at Thr305 OR Thr306 can interfere with CaM binding, but only Thr306 autophosphorylation interferes with actinin binding. However, the mechanism underlying the disruption of postsynaptic targeting by autophosphorylation at Thr305 and/or Thr306 is unclear. The final model predicts that autophosphorylation at each of these sites may modulate CaMKII-actinin interaction. The impact of the study would be significantly enhanced by additional studies using T286A, T305A, and T306A mutant CaMKIIs to determine the role of autophosphorylation at these sites on actinin-CaMKII colocalization and the spine size.

We now include experiments with neurons expressing V5-tagged CaMKIIα variants (WT, T286A, T305A, or T306A) in combination with FLAG-α-actinin-2. This has enabled us to perform antiV5/Anti-FLAG PLA to determine the effect of preventing phosphorylation at each of the three sites on association of CaMKIIα and α-actinin-2, the results of which are shown in Figure 2C and Figure 2—figure supplement 2. We found that WT CaMKIIα behaved as expected, with an ~3-fold rise in PLA puncta formation upon activation of NMDARs (Figure 2C). None of the three alanine substitution variants were markedly different from WT either before or after NMDAR activation, indicating that phosphorylation at the three sites is not required for enabling or suppressing interactions between the two proteins in this context. We note in the Results section that “T286 auto-phosphorylation is thought to be particularly important in the initial induction phase of LTP following summation of multiple Ca^2+^ transients (Yasuda et al., 2022), which is not a function that would be required using our chemical LTP protocol”. These new experiments are described in full on lines 151-177 of the Results section. We also mention the experiments later in the Results section (lines 363-364), include new methodological information related to these experiments on lines 598-602, and legends for Figure 2B and C (lines 1014-1020) Figure 2—figure supplement 2 (lines 1109-1116).

4. There is a concern about the apparent use of t-tests in all studies, even when multiple comparisons are being made in some datasets. Wouldn't ANOVAs be more appropriate?

In our original submission, we do not think there are instances where pair-wise t-test were used inappropriately. We have now taken advantage of two-way ANOVAs in the analysis of PLA data collected with WT and DCaMKII variants of GluN2B-iHA before and after NMDAR activation (Figure 6G and H). This was a powerful statistical approach to show an interaction between the presence of the CaMKII anchoring site in GluN2B subunit and receptor activation on interactions between GluN2B and both CaMKII and α-actinin-2 (see lines 404-419). We thank the reviewer to alerting us to the possibility of using ANOVAs in the analysis of our data.